# Adherent cells sustain membrane tension gradients independently of migration

Juan Manuel García-Arcos [1] ✉, Amine Mehidi[1,9], Julissa Sanchez-Velasquez[2], Pau Guillamat[1,3], Caterina Tomba [1,4], Laura Houzet[1], Laura Capolupo[5,10], Javier Espadas [1], Giovanni D'Angelo [5], Adai Colom [6,7], Elizabeth Hinde [2,8], Charlotte Aumeier [1] & Aurélien Roux [1] ✉

Tension propagates in lipid bilayers over hundreds of microns within milliseconds, seemingly precluding the formation of tension gradients. Nevertheless, plasma membrane tension gradients have been reported in migrating cells and along growing axons. Here, we show that the mechanosensitive, fluorescent membrane probe Flipper-TR visualizes membrane tension gradients in artificial and cellular membranes. Images of tension gradients allow their quantitative characterization, showing that they are long-ranged and linear in all migratory adherent cells. Using this tool, we unexpectedly reveal that tension gradients also exist in non-migrating adherent cells while they are absent in non-adherent migrating cells. This suggests that actomyosin forces can generate tension gradients even in non-moving cells, but that adhesion to a substrate is needed to sustain these gradients. Treatment of cells with drugs perturbing actomyosin show that branched actin increases tension, creating gradients. Furthermore, specific adhesion mediated by clathrin plaques colocalizes with regions of low tension, and chemical disruption of clathrin plaques strongly affect tension gradients. Altogether, our results show that the combined action of actomyosin and adhesion forces create tension gradients in the plasma membrane of adherent cells, even the ones not migrating.

Cells and organelles are separated from their environment by lipid membranes, which are self-healing fluid films[1], with unique mechanical properties. The main modulator of membrane mechanics is membrane tension, defined as the stress resulting from changing the apparent surface area of the membrane[2]. In cells, plasma membrane tension coordinates endo- and exocytosis rates both temporally and spatially[3–6], determines cell shape during single-cell migration[7–13], and collective migration[14], and regulates membrane signaling[15,16], actin polymerization[17], axon growth[18,19], and cell spreading[20,21]. Thus, membrane tension is key to coordinating cellular processes both spatially and temporally.

Interestingly, early studies on membrane tension in axons showed that tension increases from the growth cone to the cell body, contributing to the spatial regulation of cellular processes[3]. This gradient was proposed to be generated by vesicle traffic supplying membrane to the growth cone, leading to a retrograde flow of membrane.

[1]Department of Biochemistry, University of Geneva, Geneva, Switzerland. [2]School of Physics, University of Melbourne, Parkville, VIC, Australia. [3]Institute for Bioengineering of Catalonia, The Barcelona Institute for Science and Technology, Barcelona, Spain. [4]CNRS, INSA Lyon, Ecole Centrale de Lyon, Universite Claude Bernard Lyon 1, CPE Lyon, INL, UMR5270, Villeurbanne, France. [5]School of Life Sciences, Ecole Polytechnique Fédérale de Lausanne, Lausanne, Switzerland. [6]Biofisika Institute (CSIC, UPV/EHU) and Department of Biochemistry and Molecular Biology, University of the Basque Country, Leioa, Spain. [7]Ikerbasque, Basque Foundation for Science, Bilbao, Spain. [8]Department of Biochemistry and Pharmacology, University of Melbourne, Parkville, VIC, Australia. [9]Present address: Interdisciplinary Institute for Neuroscience, Université de Bordeaux, CNRS, UMR 5297, Bordeaux, France. [10]Present address: Department for Biosystems Science and Engineering (D-BSSE), ETH Zürich, Basel, Switzerland. ✉e-mail: juan.garcia@unige.ch; aurelien.roux@unige.ch

Differences in tension could be sustained by the low membrane diffusion imposed by the axon cytoskeleton[22–25]. More recent studies established that tension gradients also exist in some migrating cells[8,12].

However, how tension gradients are established in cells is still unclear[26]. If membrane tension propagates too fast, tension gradients cannot be sustained in time. In pure lipid membranes, such as giant unilamellar vesicles (GUVs), the membrane's fluidity allows tension to propagate at the speed of sound waves[27] (>0.1 m/s). As a result, tension equilibrates within tens of milliseconds, preventing the formation of stable tension gradients[28]. To observe tension gradients in vitro, lipid bilayers must adhere to a solid substrate, which reduces the diffusion of lipids[1,29]. In cells, several studies showed tension propagation to be rather fast (<1 s across the cell) and long-ranged (at the cellular scale), precluding the formation of any tension gradient[19,30–35]. Therefore, formation of tension gradients may be influenced by factors such as membrane-cortex or membrane-substrate adhesions[30,36,37] and highlight the need for a quantitative assessment to understand how tension gradients are generally established and maintained in different cellular contexts.

A common method to detect membrane tension gradients involves measuring the force required to pull membrane tethers from different parts of the cell membrane. The current understanding of membrane tension in cells has been shaped by the limitation of this technique. Tether force measurements cannot be used to assess membrane tension of the cell membrane in contact with the substrate, neither to estimate the shape of tension gradients across the cell. Moreover, tether force includes contributions not only from tension but also from membrane-cytoskeleton interactions and bilayer bending rigidity[37–42], which is lipid composition-dependent[43]. It therefore remains unclear to what extent the reported changes in tether force stem from actual membrane tension gradients, from changes in membrane-cortex binding energy, or from lipid composition changes. In this study, we aimed at visualizing membrane tension gradients, to better quantify them and explore conditions in which they could form, in particular in adherent cell membranes. membranes. For this, we used Flipper-TR, a fluorescent mechanosensitive probe, which we previously showed to report changes of membrane tension through changes of its lifetime[28]. Visualizing membrane tension with Flipper-TR has the potential to unravel the spatial organization of tension gradients over the full cell, as well as their dynamics.

In this study, we integrate model membrane systems, live-cell FLIM imaging, and pharmacological perturbations to investigate the origin and maintenance of membrane tension gradients. We show that these gradients can be sustained in non-migrating cells, provided they are adherent, thus challenging the idea that cell motility is necessary to generate them. Our findings identify actin polymerization and substrate adhesion as key regulators of these gradients, revealing a general and robust mechanism for organizing membrane tension across the plasma membrane.

## Results

### Flipper-TR reports membrane tension gradients in reconstituted membranes

To use Flipper-TR as a reporter of membrane tension gradients, we first studied in vitro the link between plasma membrane lipid composition, lipid packing, and tension gradients. The behavior of mechanosensitive probes is influenced by the lateral separation of membranes into lipid nanodomains[44–49]. Nanodomain formation is influenced by temperature, lipid composition, and mechanical factors such as the actomyosin cortex[50,51]. In vitro studies show that adhesion[52] or increased membrane tension[53–58] near the demixing transition[59] promotes lipid packing, while far from the demixing transition, increased tension leads to less lipid packing as reported by Laurdan[60,61] or Flipper-TR probes[28]. In cells, membranes appear to be close to the demixing transition[59]: increasing plasma membrane

tension by hypotonic shocks promotes nanodomain formation[62] in which lipid packing increases as reported by Flipper-TR[28], Laurdan[63], and solvatochromic probes[64]. Therefore, the response of Flipper-TR to changes in tension of the plasma membrane is expected to depend on the fact that tension can induce the lateral segregation of lipids into nanodomains[28,65].

We first tested the response of Flipper-TR in pure lipid membranes with varying compositions (both near and far from the demixing transition[66,67]) and subjected to tension gradients. We prepared glass-supported lipid bilayers (SLBs) with a defined lipid composition and stable tension gradients, as described in previous studies[1,29]. In brief, lipids were dried over large silica beads acting as carriers and placed over plasma-cleaned glass. Upon hydration, lipids wet the clean glass and expand from the bead outwards as a single bilayer. SLB expansion is counteracted by the friction of the membrane onto the glass, allowing the formation of a tension gradient (Fig. 1a)[1].

We performed confocal FLIM of the SLBs with Flipper-TR using five different lipid compositions containing Dioleoylphosphatidylcholine (DOPC), Dioleoylphosphatidylserine (DOPS), brain sphingomyelin (bSM), and cholesterol (Chol): DOPC:DOPS 60:40; DOPC:DOPS:Chol 42:28:30; DOPC:bSM:Chol 1:1:1, 1:1:2, and 2:2:1. Flipper-TR lifetime distributions qualitatively showed pronounced spatial gradients (Fig. 1b). SLBs expanded rapidly at first and slowed down following a power-law while the tension gradient relaxes (Fig. 1c)[1,29], allowing us to sample a range of tension gradients. Spatial gradients were particularly evident in fast spreading SLBs at velocities comparable to cell migration (>5μm/min) (Supplementary Video 1). The lifetime gradients were linear (Fig. 1d, $R^2 > 0.97$). This is consistent with the fact that in these expanding SLBs, the tension gradient is linear[1], and that Flipper-TR lifetime is linearly proportional to tension[28].

As expected, more ordered membranes containing sphingolipids and cholesterol led to higher Flipper-TR lifetimes (Fig. 1e). Interestingly, compositions far from the demixing transition between the homogenous and segregated states display a lower Flipper-TR lifetime at the front (leading edge) than the rear, closer to the lipid source. In contrast, compositions close to this demixing transition - meaning compositions in which phase separation can be induced - display an inverted lifetime gradient. Thus, considering the gradient from the inner to the outer part of the membrane patch, compositions close to a demixing transition (i.e., demixing compositions) displayed a positive front-rear difference while others displayed a negative front-rear difference (Fig. 1e, right panel).

Based on previous findings[28], we hypothesized that the slope inversion in demixing compositions was due to tension-induced phase separation. In this scenario, higher tension segregates the membrane into nanodomains, creating ordered lipid domains, leading to an increased Flipper-TR lifetime. To test this, we used tracer lipids. Raedler et al. showed that tracer fluorescent molecules display an exponentially decaying concentration profile in spreading SLBs with a tension gradient[1,29], with decay length depending on the molecular bulkiness of the tracer. Probes with bulky headgroups such as DOPE-atto647 localize at areas with lower lipid packing, providing visual evidence of membrane tension gradients[1]. DOPE-atto647 localized to the edge of the spreading SLBs in single-phase lipid composition, but it was excluded from the edge of SLBs composed of phase-separating compositions (Supplementary Fig. 1a). This confirmed that in spreading SLBs under tension made of demixing compositions, higher packing colocalized with areas of higher tension.

Flipper-TR displayed fluorescence intensity profiles that were inverted with respect to the DOPE-A647 profiles (Supplementary Fig. 1a). It is however more difficult to interpret these profiles in terms of concentration since the quantum yield of Flipper-TR increases with lipid packing[68]. But the higher Flipper-TR fluorescence intensity at the

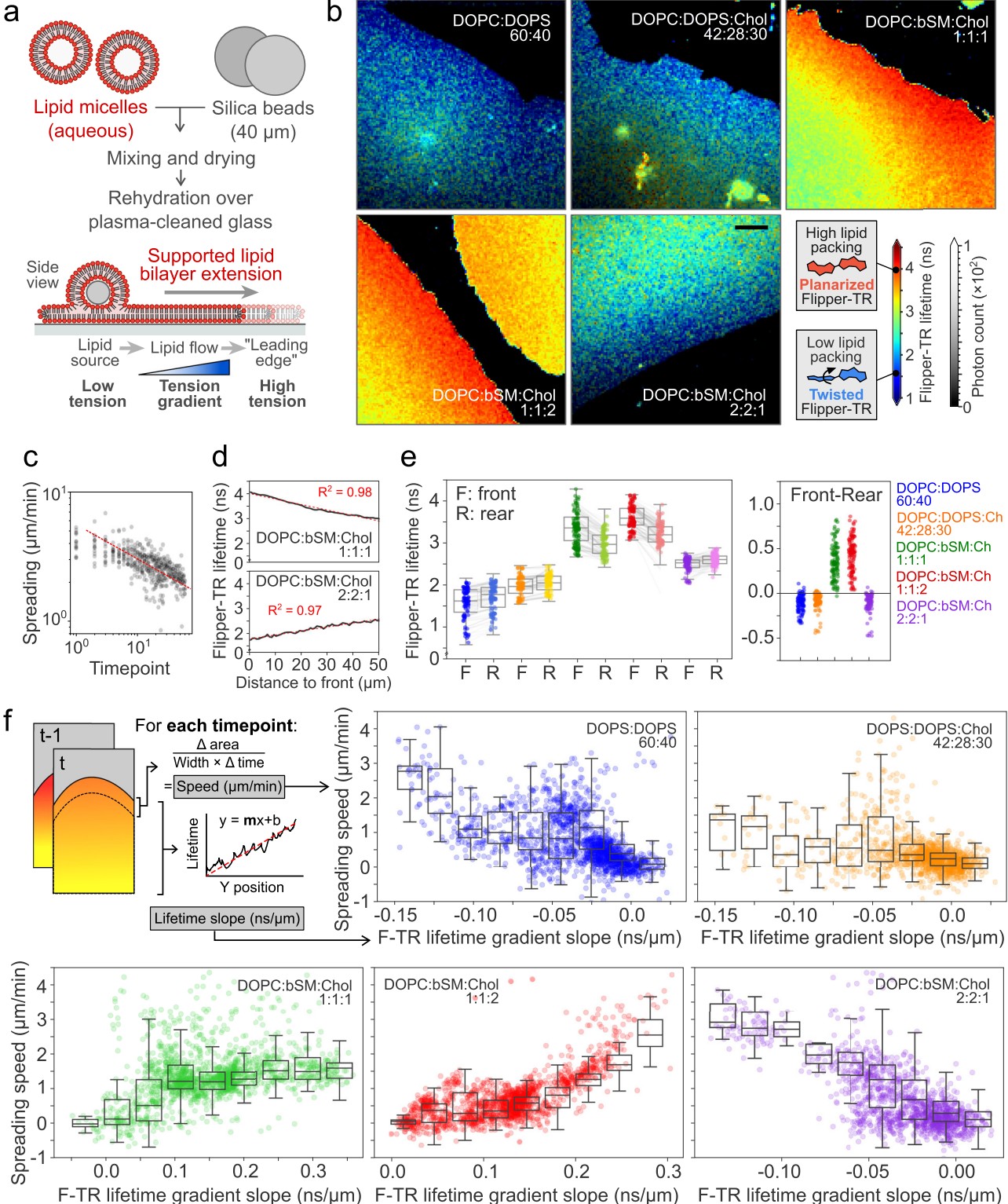

**Fig. 1 | Flipper-TR reports membrane tension gradients in reconstituted membranes. a** Schematics of supported lipid bilayer experiments. The bilayer wets the glass and expands from the lipid source due to tension-induced flows. The spreading velocity is proportional to the tension gradient. **b** Examples of expanding supported lipid bilayers with different compositions. Color indicates average Flipper-TR fluorescence lifetime (ns); black mask shows the average photon count. Scale bar, 10 μm. **c** Bilayer spreading speed (μm/min) over time in log-log plot. Red line for visual reference. **d** Average Flipper-TR lifetime (ns) as a function of the distance from the leading edge, for two representative examples of two

compositions. **e** Front (expanding edge) and rear (closest to lipid source) average Flipper-TR fluorescence lifetime in supported lipid bilayers with different compositions. Lines represent data pairing. Right, front-rear differences for different lipid compositions. **f** Top left, diagram of the quantification. Bilayer spreading speed (μm/min) as a function of linear fits of spatial Flipper-TR gradients (ns/μm) for different lipid compositions. Histograms of binned data overlaid in black. $N = 5$, $n = 277$. Boxplots show the median (center line), 25th and 75th percentiles (bounds of the box), and whiskers extending to the most extreme data points within $1.5 \times$ the interquartile range from the box.

edge of SLBs with demixing compositions is consistent with the presence of higher packing in edge regions.

Qualitatively, only bilayers that are expanding – thereby having a tension gradient – displayed lifetime gradients, supporting that lifetime gradients originated from tension gradients. To better understand the interplay between Flipper-TR lifetime gradients and membrane tension, we plotted the slope of gradients as a function of the SLB expansion speed. For long time points, the speed reduced to zero as well as the slope value, confirming that when SLBs stop spreading and dissipate their tension gradient, the Flipper-TR lifetime gradient also vanishes (Supplementary Video 1). For SLBs with compositions far from the demixing transition, the slope of the Flipper-TR lifetime gradient (see Methods) increased from negative values to zero roughly proportionally to expansion speed (Supplementary Fig. 1b, DOPC:DOPS 60:40 and DOPC:bSM:Chol 2:2:1). For SLBs with compositions close to the demixing transition, the slope decreased instead from positive values to zero, also roughly proportionally to expansion speed (Supplementary Fig. 1b, DOPC:bSM:Chol 1:1:1 and 1:1:2). This effect is robust across a wide range of expansion speeds and lifetime gradients (Fig. 1f).

Overall, these data showed that Flipper-TR can report tension gradients in model membranes, but that the slope of the gradient strongly depended on the lipid composition and the spreading forces.

## Flipper-TR reveals tension gradients in migrating cells

In migrating cells, membrane tension gradients were first evidenced in fish keratocytes[12], which have a crescent shape and persistent, fast migration driven by a large lamellipodium at the leading edge[10]. Since Flipper-TR can report tension gradients, we aimed to test whether it displayed a lifetime gradient during keratocyte migration. Keratocytes were labeled with Flipper-TR and their bottom plasma membrane (in contact with the substrate) was imaged using confocal FLIM (see Methods)[69]. Our results confirmed that cells robustly display large lifetime differences between the lamellipodium leading edge and the cell rear (Fig. 2a) (distribution reported as mean ± s.d.): Lifetime$_{front}$ = 4.8 ± 0.13 ns; Lifetime$_{rear}$ = 4.65 ± 0.16 ns; ΔLifetime = 0.20 ± 0.12 ns. This is consistent with the literature, reporting higher lipid packing and higher tension values at the front of migrating cells[12]. At the level of single cells, front and rear values fluctuate over time while keeping the gradient in place (Fig. 2b).

Tension gradients might vary across cell lines and depend on their migratory behaviors. We therefore selected cell lines to cover a wide range of migratory behaviors: U2OS and RPE1, which migrate fast with a persistent direction, and HeLa and Cos7 which appear slow and less persistent (Supplementary Fig. 2a-b). Qualitatively, as for keratocytes, the cells showed large heterogeneities of Flipper-TR lifetime (Fig. 2c), being higher at cell edges and decaying toward the cell center. Contrary to keratocytes, these cell lines have overall a less persistent migration, allowing us to manually sort cells into polarized and non-polarized categories based on their shape. While cells displaying a highly polarized shape showed clear differences between front and rear spatial decays (Fig. 2c, d), non-polarized cells did not (Supplementary Fig. 2c, d). In contrast to keratocytes, the lifetime gradient is aligned with the cell direction and not always with the long axis of the cell shape. Lifetime values decayed linearly over more than 10 μm from the cell leading edge, while for the two faster migrating cells they decayed exponentially within a few microns to the baseline levels from the rear (Fig. 2d, e). Lifetime differences between front and rear were more pronounced in cells with elongated morphologies (Fig. 2f). Overall, the cell lines that displayed a more persistent migration were the ones that displayed the most robust Flipper-TR gradients. Interestingly, U2OS cells displayed overall lower lifetime values than other cell types but had the same gradient as the other persistent migratory cell type, RPE1 (Fig. 2e). Since plasma membrane lipidomes are cell-type specific, offsets in the average lifetime may be due to a specific

lipid composition in this cell line[70]. We concluded from these data that the migratory leading edge can establish long-range linear lifetime gradients with higher tension at the edge. As lifetime gradients of Flipper-TR report tension gradients in SLBs (Fig. 1), our data supports that Flipper-TR visualizes membrane tension gradients in migratory cells.

Since the lowest lifetimes often correspond to the central region of the cells, coinciding with the nucleus position, we tested whether the presence of the nucleus could influence the reported plasma membrane Flipper-TR lifetime gradients. For this, we characterized the lifetime of the bottom membrane of enucleated cells (Methods)[71]. Enucleated U2OS and RPE1 cells maintained similar Flipper-TR lifetime distributions (Supplementary Fig. 2e-g) implying that the nucleus is not generating lifetime gradients within the plasma membrane.

As actomyosin dynamics is the primary force-generating process that drives cell migration, actin could contribute to the gradients. We therefore studied the colocalization of the actin cytoskeleton with Flipper-TR lifetime gradients. For this, following Flipper-TR lifetime imaging, U2OS cells were fixed and labeled for actin and vinculin (see Methods). In polarized cells, actin and vinculin had a stereotypical localization pattern composed of a dense, narrow actin strip marking the lamellipodium edge and a less dense actin region with small elongated focal adhesions corresponding to the lamella. At the rear, we observed less cortical actin and thick stress fibers originating from focal adhesions (Fig. 2g). Regions of highest and lowest lifetime coincided with characteristic features in the actin organization. Flipper-TR lifetime increased both in the lamellipodium and lamella regions (Fig. 2h). We used actin images to train a machine learning algorithm using Ilastik[72] to segment cells into lamellipodium, lamella, and cell body regions, and calculate their average Flipper-TR lifetime values (Fig. 2i). This confirmed that the spatial differences in actin cytoskeleton morphology led to different Flipper-TR lifetimes.

Actin polymerization extends the leading edge of lamellipodium and thereby could increase plasma membrane tension[13,20,31,73]. To study this, we performed live FLIM on RPE1 and U2OS cells, to track cell edge dynamics simultaneously with Flipper-TR lifetime. FLIM time-lapses on migrating cells confirmed that, in the event of cell repolarization, the region of high lifetime dynamically changes localization to the new leading edge (Supplementary Video 2). For every point of the cell edge, the gradient of Flipper-TR lifetime (from the cell edge inwards) and the edge velocity were measured in RPE1 and U2OS cells. Plotting the spatial lifetime gradient binned by edge velocity revealed that the protrusion speed is positively correlated with increased Flipper-TR lifetime (Fig. 2j). The protruding parts of the cell display higher Flipper-TR lifetime values than the retracting ones (Fig. 2k), as follows (mean ± s.d.): U2OS protrusions = (4.63 ± 0.16 ns); U2OS retractions = (4.46 ± 0.21 ns); RPE1 protrusions = (4.76 ± 0.36 ns); RPE1 retractions = (4.47 ± 0.22 ns). These results confirm a temporal correlation between membrane protrusion and increased Flipper-TR lifetime.

Analogous to the analysis based on cell shape only (Fig. 2d), protruding regions displayed a long-range linear lifetime decay. Interestingly, retracting regions, where membrane tension is expected to be lower, displayed a short-range increase in Flipper-TR lifetime values at the cell edge, decaying rapidly over 1–2 μm. This counterintuitive signal led us to investigate whether membrane–substrate interaction could influence lifetime values independently of actin-driven protrusion. Across all cell types, we found that the membrane in close contact with the glass always exhibited lower lifetime values than the rest of the cell (Supplementary Fig. 3a). To rule out that this top-bottom lifetime difference was due to optical artefacts, cells were imaged from the side by growing them on microstructured surfaces onto which the basal surface was oriented 120° to the focal plane (Supplementary Fig. 3b-c). These images confirmed that the adhesion patch of cells had a lower lifetime. Surprisingly, GUVs adhered to PEG-

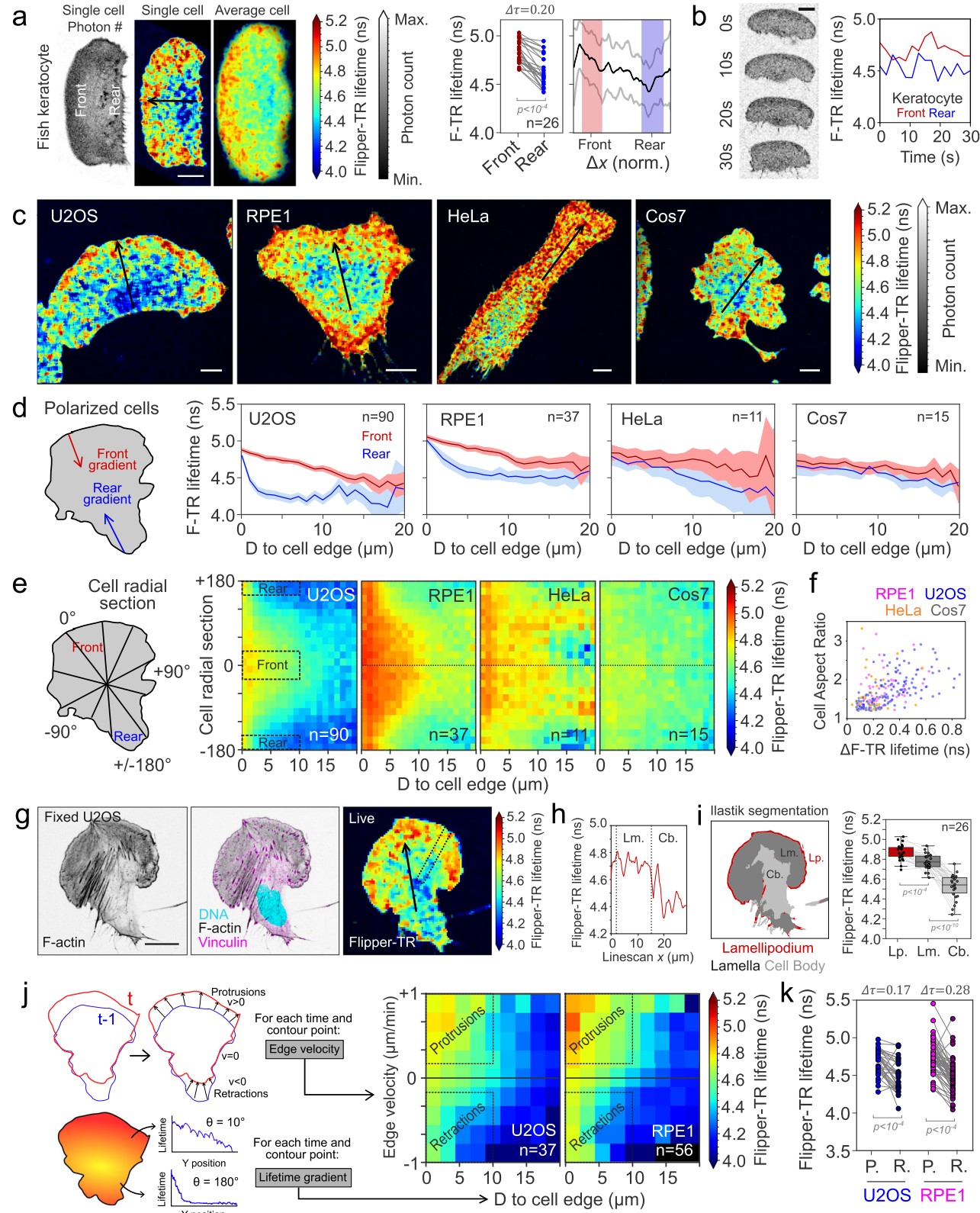

g-PEG-coated coverslips displayed similar lower lifetime in the adhesion patch, particularly liquid-ordered compositions (Supplementary Fig. 3d-e). In contrast, membrane regions not bound to the substrate such as edges had higher lifetimes, similar to the cell top membrane (see decays on Supplementary Fig. 3d). A recent study using Laurdan imaging reported analogous findings, with lower GP in the adhesion patch of both cells and GUVs[74]. The authors interpreted this change as

a flattening of small fluctuations in the adhesion patch that renders the membrane lipids less packed. Their interpretation supports that the rim of higher lifetime circling cells and GUVs corresponds to the unbound parts of the membrane that connect to the top part, still imaged in the confocal slice of the image.

From these data, we concluded that Flipper-TR visualizes tension gradients in migrating cells, supporting that it can be used to study the

**Fig. 2 | Leading-edge extension increases Flipper-TR lifetime in migrating cells.**
**a** Left, representative live confocal FLIM image of fish keratocytes labeled with
Flipper-TR. Color indicates average Flipper-TR fluorescence lifetime (ns); black
mask shows average photon count. Right, average Flipper-TR fluorescence lifetime
at the front and rear 20% areas of migrating fish keratocytes ($N = 4$, $n = 27$, Welch's
$P < 10^{-4}$), with lifetime profile along the normalized position (mean ± standard
deviation). Front and rear regions are shaded in red and blue, respectively. **b** Left,
representative time-lapse image of migrating fish keratocytes. Right, average
Flipper-TR fluorescence lifetime at the front (red) and rear (blue) 20% of cell area
during the time lapse. **c** Representative confocal FLIM image at the bottom plane of
polarized U2OS, RPE1, HeLa, and Cos7 cells on glass, labeled with Flipper-TR. Color
indicates average Flipper-TR fluorescence lifetime (ns); black mask shows average
photon count. **d** Average Flipper-TR fluorescence lifetime as a function of D, dis-
tance from the edge at the front (red) and rear (blue) of polarized U2OS ($n = 90$),
RPE1 ($n = 37$), HeLa ($n = 11$), and Cos7 ($n = 15$) cells. Line represents mean ± standard
deviation. **e** Average Flipper-TR fluorescence lifetime as a function of D, distance
from the edge and radial position (front at 0°) of polarized U2OS ($n = 90$), RPE1
($n = 37$), HeLa ($n = 11$), and Cos7 ($n = 15$) cells. Color indicates average Flipper-TR

fluorescence lifetime (ns). **f** Difference of average Flipper-TR fluorescence lifetime
between front and rear 10-μm regions as a function of cell aspect ratio for U2OS
(blue, $n = 132$), RPE1 (magenta, $n = 43$), HeLa (orange, $n = 63$), and Cos7 (gray, $n = 32$)
cells. **g** Left, representative image of fixed U2OS cells labeled with phalloidin-
Alexa647, Hoechst, and anti-vinculin-alexa488. Right, live image of U2OS labeled
with Flipper-TR. Dotted region indicates line scan region for panel **h**. **h** Average
Flipper-TR fluorescence lifetime along a line scan marked in panel **g**. Dotted lines
indicate lamellipodium, lamella (Lm.), and cell body (Cb.). **i** Left, segmentation of
the representative cell in panel G into lamellipodium (red, Lp.), lamella (dark gray,
Lm.), and cell body (light gray, Cb.). Right, average Flipper-TR fluorescence lifetime
in these regions ($n = 26$, paired Welch's P). Boxplots show the median (center line),
25th and 75th percentiles (bounds of the box), and whiskers extending to the most
extreme data points within 1.5 × the interquartile range from the box. **j** Left, diagram
of the quantification. Right, average Flipper-TR fluorescence lifetime as a function
of D, distance from the edge and edge velocity in U2OS ($n = 37$), RPE1 ($n = 56$) cells.
Color indicates average Flipper-TR fluorescence lifetime (ns). **k** Average Flipper-TR
fluorescence lifetime at protruding (P) and retracting (R) regions in U2OS ($n = 37$),
RPE1 ($n = 56$) cells (paired Welch's P). **a–c**, **g**: Scale bar, 10 μm.

dynamics and the organization of membrane tension gradients in
different cellular contexts.

## Membrane tension gradients are present in adherent, non-migrating cells

After confirming that Flipper-TR can reveal membrane tension gra-
dients in expanding membranes both in migrating cells and in vitro, we
investigated if plasma membrane tension gradients can be present in
non-migrating cells. To this end, we cultured single HeLa cells on
fibronectin-coated micropatterns[75,76] and labeled them with Flipper-
TR. Controlling cell shape allowed us to acquired Flipper-TR lifetime
images of cells with comparable spatial organization of intracellular
components, including nuclei, actin cytoskeleton, and focal adhesions
(Fig. 3a-b). In single cells, Flipper-TR lifetime showed a heterogeneous
spatial distribution at the bottom plasma membrane (Fig. 3c), as pre-
viously observed[5], while its distribution was homogeneous on the non-
adhered, upper membrane (Supplementary Fig. 4a). Different cell
shapes display no significant differences in the average Flipper-TR
lifetime, neither the bottom (Fig. 3d) nor upper planes (Supplementary
Fig. 4b). As reported earlier for non-micropatterned cells, the average
Flipper-TR lifetime values were different between the upper and bot-
tom planes (Supplementary Fig. 4c), (mean, s.d.): Lifetime upper =
$(4.98 \pm 0.05$ ns); Lifetime bottom = $(4.71 \pm 0.06$ ns), $\Delta$Lifetime =
$(0.26 \pm 0.07$ ns).

The standard cell shapes allowed us to build spatial probability
maps of Flipper-TR lifetime (Fig. 3e). The average lifetime at the bot-
tom membrane was comparable between adhesive and non-adhesive
regions (Fig. 3f). However, membrane protrusions escaping the
micropattern, detaching from the non-adhesive coating of the surface,
exhibited higher lifetime values (Fig. 3f). We interpreted these as
arising from membrane dewetting as discussed above, and focused on
the gradients found across the bottom surface. The spatial probability
maps of Flipper-TR lifetime suggested that the gradients across the cell
or along the cell contour (Fig. 3g) are rather set by the position of the
cell edge and the micropattern. We further analyzed this by assigning
two spatial descriptors to each pixel, namely the shortest distance to
the cell edge and the shortest distance to the micropattern boundary
(negative distances account for pixels inside). By plotting the average
lifetime as a function of these two distances we can compare the life-
time gradients we obtained from the three different pattern shapes
(Fig. 3h). The fact that the gradients exhibited the same distribution
regardless of the shape confirmed that the positions of the cell edge
and the micropattern boundary are main determinants of the Flipper-
TR lifetime distribution (Fig. 3h, average panel).

We next studied how changing the adhesive area of the micro-
pattern would change the gradients. In larger patterns where HeLa

cells are more stretched, additional protrusions were suppressed,
decreasing the lifetime in the middle of the ring pattern. In smaller
patterns, the effect was the opposite, increasing the lifetime in the
middle of the ring pattern (Supplementary Fig. 4d-e). On pattern with
wider adhesion rings, the lifetime decreased at the inner boundary of
the pattern (Supplementary Fig. 4f). In full disc patterns, lifetime was
distributed homogeneously (Supplementary Fig. 4f). In all cases, the
increase of Flipper-TR lifetime qualitatively correlated with regions
with protrusions, and a decrease of lifetime at the inner boundary of
the ring. Therefore, the total adhesive area - whether because the
pattern is bigger or contains more adhesive parts - does not seem to
strongly affect the gradient formation (Supplementary Fig. 4f).

We then looked if different types of cells displayed different
lifetime gradients on micropatterns. We compared Hela to RPE1
cells as they displayed the most different gradients and migratory
behavior (Fig. 1). Similar to HeLa, Flipper-TR lifetime in RPE1 cells
decreased from the cell edges to the cell center, but the increase
of lifetime at plasma membrane regions over non-adhesive
regions was absent (Supplementary Fig. 4g). We attribute this
increase to protrusive activity and ruffle formation over non-
adhesive areas in HeLa (Supplementary Video 3), which are not
present in RPE1 cells. This suggested that cells with different
adhesion and actin organization displayed different lifetime
gradients.

In support of this, and despite the standardization of cell shape,
cells still exhibit significant differences in vinculin and actin organi-
zation. "Low protrusion" cells display a very dense vinculin recruitment
on focal adhesions, thick stress fibers, and lack protrusions. "High
protrusion" cells display homogeneous vinculin localization, no stress
fibers, and numerous protrusions projecting outwards (Supplemen-
tary Fig. 5a). By comparing the lifetime gradients of the 10% cells with
the highest protrusion area and the 10% cells with the lowest, we found
that cells with more protrusive behavior had higher overall lifetime
values and vice versa (Supplementary Fig. 5b-d). This confirms, as in
migrating cells, that protrusions play a fundamental role in membrane
tension in patterned cells.

Overall, these results showed that membrane tension gradients
appear in non-migrating, adherent cells, and that cell edges and
adhesive structures set the gradient boundaries.

## Branched actin increases membrane tension

Membrane protrusions can be driven by linear actin filaments and
branched actin networks. Each of these interacts with the membrane
differently, particularly in how they transmit forces. In this context,
branched actin is thought to have a unique role in increasing mem-
brane tension[77–79]. To study the molecular mechanisms setting the

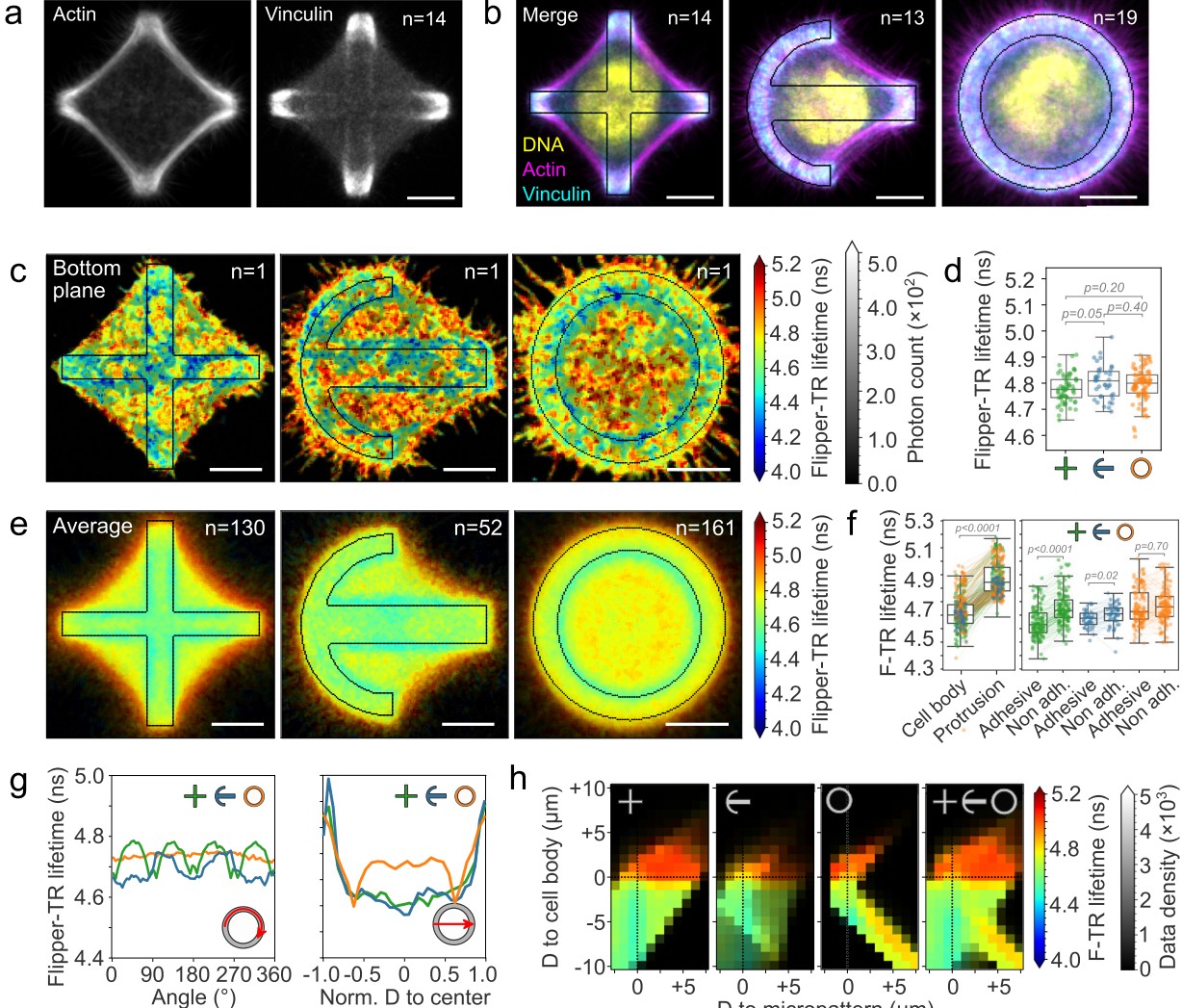

**Fig. 3 | Membrane tension gradients are shaped by cellular organization.**
**a** Average fluorescence images at bottom plane of cross-micropatterned HeLa cells stained with phalloidin for actin and anti-vinculin antibody. **b** Color overlay of Hoechst (DNA, yellow), vinculin (focal adhesions, cyan), and phalloidin (F-actin, magenta) of micropatterned HeLa cells with different shapes. **c** Representative confocal FLIM images at the bottom plane of HeLa cells labeled with Flipper-TR on cross, crossbow, and ring micropatterns. **d** Average Flipper-TR fluorescence lifetime at the bottom plane of cross, crossbow, and ring micropatterned cells ($n = 130$, 52, 161, Welch's P). Data distribution in black. **e** Average confocal FLIM images at the bottom plane of HeLa cells labeled with Flipper-TR on cross, crossbow, and ring micropatterns. **f** Average Flipper-TR fluorescence lifetime (ns) in cell regions of cross (green), crossbow (blue), and ring (orange) micropatterned cells. Data distribution in black. Gray, Welch's P value. **g** Top, average Flipper-TR fluorescence lifetime along the cell contour of cross (green), crossbow (blue), and ring (orange) micropatterned cells. Bottom, lifetime distribution across the cell. **h** Average Flipper-TR fluorescence lifetime as a function of D, distance from micropattern edge (negative: inside of the micropattern, positive: outside) and cell body edge (negative: inside of the cell body, positive: protrusions). Color indicates average Flipper-TR fluorescence lifetime (ns); black mask shows data point density. **a–c, e**: Scale bar, 10 μm. **d,f**: Boxplots show the median (center line), 25th and 75th percentiles (bounds of the box), and whiskers extending to the most extreme data points within 1.5 × the interquartile range from the box.

tension gradients, we generated maps of Flipper-TR lifetime under several pharmacological treatments: (i) 1 μM Latrunculin A ('LatA') for inhibiting actin polymerization, (ii) 85 μM CK-666 for inhibiting Arp2/3-mediated branched actin nucleation, (iii) 10 μM NSC668394 ('NSC66') for inhibiting ezrin – a major actin-membrane linker protein, (iv) 20 μM Y-27632 for inhibiting Rho kinase, responsible for myosin contractility, and (v) the JLY cocktail consisting of 20 μM Y-27632, 5 μM Latrunculin B, and 8 μM Jasplakinolide to block actin dynamics[80]. JLY-treated cells were taken as a zero baseline to decouple the apicobasal gradient from the gradients generated by actin activity. For each treatment and each shape (cross and crossbow shapes are shown in Supplementary Fig. 6), we imaged the actin cytoskeleton and focal adhesions (Fig. 4a) and quantified: (i) the raw spatial lifetime distribution (Fig. 4b), (ii) the spatial lifetime distribution using JLY as a reference (Fig. 4c, lifetime images from 4b were subtracted with the JLY image), (iii) the average

lifetime (Fig. 4d), and (iv) the maximal amplitude of the gradients (Fig. 4e).

While the top-bottom lifetime difference was conserved in all drug treatments (Supplementary Fig. 6e), the lifetime distribution in the bottom plane displayed strong changes. Inhibiting actin polymerization by LatA and JLY had a comparable effect on the Flipper-TR lifetime map (Fig. 4a-c). Inhibiting actin polymerization abolished protrusions and thus flattened tension gradients in non-adhesive areas, but not the short gradients at cell edges, as they are due to the top-bottom lifetime difference. In order to remove this gradient at the rim, and better observed changes of tension generated by drug treatments, we subtracted the JLY from the other conditions to visualize only the lifetime gradients generated by active actomyosin (Fig. 4c). These differential images showed that LatA and JLY are very similar, with no visible gradients, apart from the one at the rim that

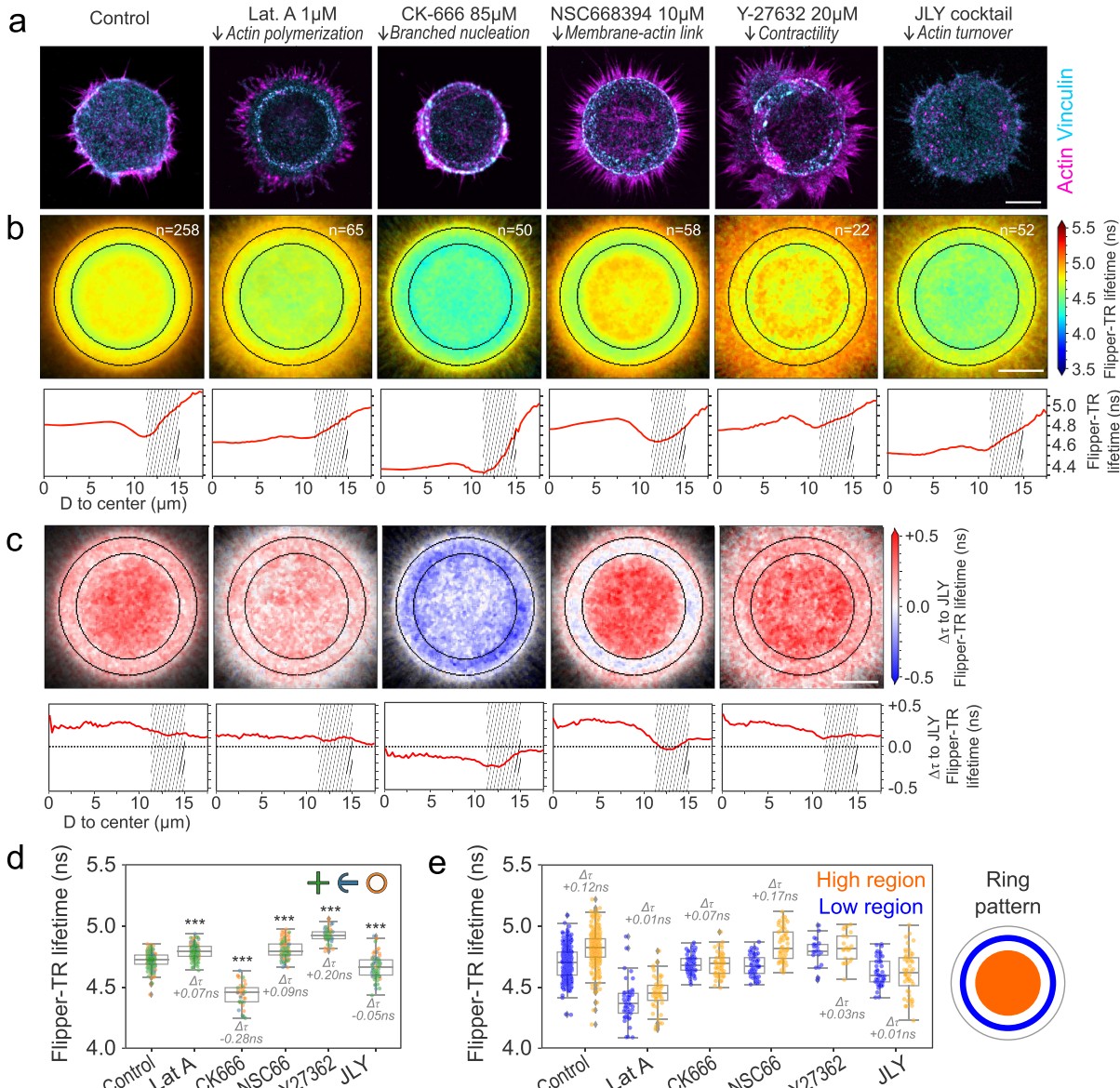

**Fig. 4 | Branched actin increases membrane tension. a** Representative fluorescence images at bottom plane of HeLa cells on ring micropatterns stained with phalloidin (magenta) and vinculin antibody (cyan) under different drug treatments. **b** Top, average confocal FLIM images at the bottom plane of HeLa cells labeled with Flipper-TR under different drug treatments. Bottom, average Flipper-TR lifetime as a function of the distance from center of micropattern. Shaded areas represent adhesive micropattern location. **c** Top, flipper-TR lifetime difference respective to JLY treatment. Bottom, flipper-TR lifetime difference respective to JLY treatment as a function of the distance from center of micropattern. **d** Average Flipper-TR

lifetime (ns) per cell of cross (green), crossbow (blue), and ring (orange) micropatterned cells under different drug treatments. Labels represent Welch's test $P$ value ($P < 10^{-3}$) and difference from control in ns. Data distribution in black. **e** Quantification of flipper-TR lifetime gradient: average lifetime in 'high region' (center of the micropattern) and 'low region' (inner edge of micropattern) under different drug treatments. **a–c**: Scale bar, 10 μm. **d,e** $N$ as marked per condition in panel **b**. Boxplots show the median (center line), 25th and 75th percentiles (bounds of the box), and whiskers extending to the most extreme data points within $1.5 \times$ the interquartile range from the box.

comes from the top-bottom lifetime difference. Non-treated cells show a pronounced inward gradient (Fig. 4e), linearly increasing in the site of protrusions (inner, non-adhesive part of the ring pattern). This confirmed that the linear gradients observed earlier in the lamellipodial regions of migrating cells and protruding regions (Fig. 2) generally depend on actin polymerization.

Cortical F-actin in HeLa cells is composed about half-half of branched (Arp2/3-based) and linear (formin-based) networks[81]. The nucleation of cortical actin is a dynamic process with a lifetime of about a minute, where nucleators compete for available G-actin. Arp2/3 inhibition led to the formation of more filopodia, less lamellipodia, and thicker focal adhesions and stress fibers (Fig. 4a). Surprisingly, the inhibition of Arp2/3 reduced considerably the average

Flipper-TR lifetime value, more pronounced than LatA or JLY treatments (Fig. 4b–d), resulting in a dimmer gradient than in control conditions (Fig. 4e). This suggests that, in control conditions, lamellipodial extension – promoted by Arp2/3 branching – is generating higher membrane tension, setting a longer characteristic decay length of the gradient. The specific role of branched actin networks was also supported by Rho kinase inhibitor Y-27632 treatment, which significantly extended lamellipodia (Fig. 4a). This resulted in a global increase in the average lifetime (Fig. 4d) and reduced the amplitude of the lifetime gradients (Fig. 4e).

Finally, to inhibit membrane-to-cortex attachment, we used NSC668394, which inhibits ezrin activity by preventing its T567 phosphorylation, a major membrane-cortex linker in HeLa cells[82]. Cells

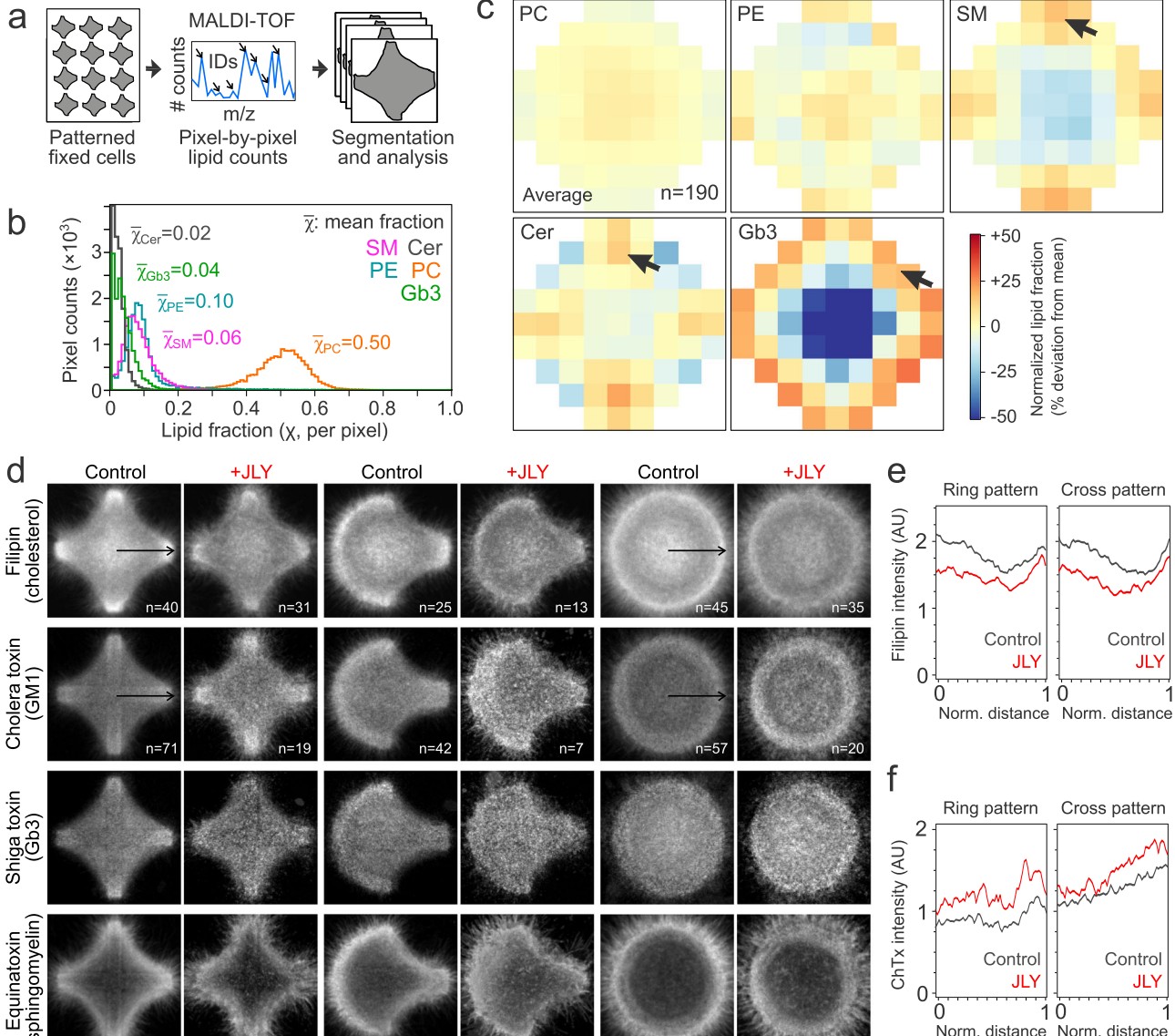

**Fig. 5 | Spatial distribution of lipid composition in patterned cells. a** Schematics representing MALDI-MSI procedure. **b** Histogram of pixel-wise lipid fractions, and average lipid fractions for different lipid species: sphingomyelin (SM, magenta), ceramide (Cer, gray), phosphatidylethanolamine (PE, blue), phosphatidylcholine (PC, orange), and globotriaosylceramide (Gb3, green). **c** Cell averaging of spatial MALDI-MSI of different lipid species. Color scale represents local differences from mean value normalized in terms of standard deviations. Arrows indicate observed lipid accumulation. **d** Average fluorescence images at basal plane of HeLa cells in cross, crossbow, and ring ($n > 20$ each) micropatterns stained with filipin, Cholera toxin, Equina toxin, and Shiga toxin targeting cholesterol, GM1, sphingomyelin, and globotriaosylceramide, respectively, under control and JLY treatment. Scale bar, 10 μm. **e, f** Average fluorescence intensity profile of filipin (B, cholesterol) and Cholera toxin (C, GM1) across the line marked in panel D for control and JLY treatment.

under ezrin inhibition displayed similar protrusive morphology as the control cells but displayed smaller focal adhesions with less vinculin (Fig. 4a), resulting in a stronger gradient (Fig. 4c). This indicates that with less membrane-to-cortex attachment, cells can still increase membrane tension at protrusions, but this tension dissipates faster away from protrusive regions. We, therefore, concluded that actin-membrane attachment may be involved in reducing the local diffusion of membrane components, thereby reducing tension propagation and gradients. Overall, these results confirm that tension generation is controlled by the dynamics of the actin cytoskeleton, specifically of branched actin, via membrane-to-cortex attachment.

## Distribution of main lipid components is independent from tension gradients

We questioned to what extent the Flipper-TR lifetime gradients observed in patterned cells can be explained by spatial variations in lipid composition, since Flipper-TR can be sensitive to composition and not only tension. To investigate this, we analyzed the spatial distribution of lipid composition by using Matrix-Assisted Laser Desorption Ionization Mass Spectrometry Imaging (MALDI-MSI). This technique allows the mass-spectrometry analysis of the chemical content of each pixel in an image providing a detailed analysis of most lipid species with a subcellular resolution[83]. The subcellular resolution was achieved by averaging over many images thanks to standardized cell shapes (Fig. 5a and Methods).

We measured the fraction of different lipid species in cross-shaped micropatterned cells: namely phosphatidylcholine (PC), phosphatidylethanolamine (PE), ceramide (Cer), sphingomyelin (SM), and globotriaosylceramide (Gb3) (Fig. 5b). From our results on model membranes and the literature[1], we expect that composition gradients can be directly linked to tension gradients only in lipids in trace amounts with extreme bulkiness (for example DOPE-atto647 in

Supplementary Fig. 1a). Consistent with our hypothesis, lipid species that are abundant in these membranes (PC, PE) display homogeneous distributions while less abundant lipid species (Cer, SM, and Gb3) display heterogeneous spatial distributions in the form of lipid concentration gradients (Fig. 5c). The spatial gradient of Gb3, a rare and bulky lipid species which accumulates at protrusions, is consistent with the tension gradient (Fig. 3e), maximal at the cell edges. Interestingly, the ceramide and sphingomyelin distribution showed a moderate preference for focal adhesion sites which do not correlate with an increase in Flipper-TR lifetime (Fig. 5c). This supports that the Flipper-TR lifetime differences are not due to differences in lipid composition, as we could expect sphingolipids to increase membrane order and Flipper-TR lifetime. To further explore the spatial distribution of lipid composition in micropatterned cells and complement some of the limitations of MALDI-MSI, we used fluorescent dyes, Filipin, Cholera toxin, Equina toxin, and Shiga toxin that specifically bind to cholesterol, GM1, sphingomyelin, and Gb3 lipids, respectively (Fig. 5d). We also imaged JLY-treated cells on the bottom plane after fixation and staining. Qualitatively, in agreement with MALDI data, the concentration of all these lipid species tends to increase at the vertices of the cells on cross-shaped patterns, but the lipid packing effects for sphingolipids associated with lipid rafts[84] cannot explain the spatial variations observed in the average lifetime maps. Importantly, JLY treatment does not change the spatial distribution of cholesterol or other lipid species contributing to lipid order.

For example, the increase in cholesterol observed in the middle of ring patterns is not abolished with JLY, contrary to the lifetime, comforting our conclusions. Taken together, these data support that variations in lipid composition do not contribute to the observed Flipper-TR lifetime gradients in patterned cells.

## Lipid diffusion and flows are limited in patterned cells

Previous studies have shown that gradients in membrane tension are associated with lipids flowing[18] from low to higher tension regions of neuronal growth cones. On the other hand, other studies on migrating cells have shown either no flow[85] or rearward flow[86] in the reference frame of the cell. To explore if tension and flows are coupled in our system, we analyze the spatial distribution of lipid diffusion and flows in micropatterned cells, in control and JLY treatment, which completely abolished actin flows (Supplementary Fig. 7a, b). Punctual Fluorescence Recovery After Photobleaching (FRAP) on membrane dyes assays revealed no significant difference in the recovery times across different cell regions at the bottom membrane (Supplementary Fig. 7c), note that FRAP $t_{1/2}$ does not change on the adhesive and non-adhesive areas of the edges). We further tested the presence of flows using FRAP, bleaching a line tangential to the cell edge and tracking if the fluorescence minima moved during recovery. These assays showed that intensity minima moved towards the cell edge (Supplementary Fig. 7d). FRAP data can be interpreted as resulting from either a bottom plasma membrane flow directed towards the cell edges that is non-dependent on actin dynamics, or from the existence of a passive diffusion barrier at the leading edge[87], which could limit fluorescence recovery inwards. Moreover, this apparent lipid flow did not change in the absence of actin dynamics (Supplementary Fig. 7d, JLY treatment). We aimed at discriminating between lipid flows and diffusion barriers with other means than bleaching.

For this, we used fluorescence fluctuation spectroscopy (FFS) and spatiotemporally studied lipid dynamics without photobleaching by autocorrelation function (ACF) and pair correlation function (pCF) analysis across line scan FFS measurements[88–90]. Single-channel line scans across the bottom membrane of patterned cells, stained with the plasma membrane dye CellMask, were acquired at a rate faster than diffusion (Supplementary Fig. 7e,f). Then the local versus the long-range mobility of the diffusing dye molecules across the membrane was extracted by an ACF and pCF analysis of the acquired fluorescence

fluctuations with themselves (autocorrelation) versus at specific distances in the forward or reverse direction (cross-correlation) (Supplementary Fig. 7g, see Methods). We positioned the line scans crossing the pattern boundaries, where sharp decrease of the Flipper-TR fluorescence lifetime was observed (Supplementary Fig. 7h). The ACF analysis showed that lipid mobility does not change between the adhesive and non-adhesive areas (Supplementary Fig. 7j). Our diffusion values are consistent with published point FCS measurements of the same lipid probe ($\langle D_{cross\ control} \rangle = 2.2 \pm 2.7 \mu m^2/s$; $\langle D_{ring\ control} \rangle = 1.9 \pm 2.6 \mu m^2/s)^{91}$. Under JLY treatment, we noticed a weak trend decreasing overall diffusion and transport ($\langle D_{cross\ JLY} \rangle = 1.7 \pm 2.0 \mu m^2/s$; $\langle D_{ring\ JLY} \rangle = 1.5 \pm 2.0 \mu m^2/s$). The pCF analysis revealed an absence of membrane lipid flow (Supplementary Fig. 7i, no significant differences between forward and retrograde transport).

From these data, we concluded that patterned cells do not display a steady-state lipid flow at their bottom membrane, despite the presence of membrane tension gradients. This is consistent with the fact that membrane tension gradients are maintained through time, as they are not dissipated by lipid flows.

## Substrate rigidity and adhesiveness modulate membrane tension gradients

Our results in patterned cells indicate so far that tension gradients can exist in non-migrating cells due to adhesion patterns and actin polymerization. We then questioned whether they were exclusive to adherent cells, and what are the conditions for gradients to emerge. To address this, we plated HeLa cells in fibronectin-coated substrates of varying rigidity (Fig. 6a). In fibronectin-coated soft polyacrylamide (PAA) hydrogels of increasing rigidity, cells extended protrusions to cover a larger area, and a lateral Flipper-TR gradient emerged (Fig. 6b). Interestingly, the top-bottom lifetime difference was only present in cells plated on very rigid substrates, such as PDMS or glass, in HeLa (Fig. 6c) or any other cell line we have tested (see Supplementary Fig. 8c for U2OS and RPE1). Interestingly, soft (0.1 kPa) and hard (50 kPa) hydrogels display the same top-bottom lifetime difference while maintaining very different lateral lifetime gradients. In other words, lateral lifetime gradients form independently of the top-bottom lifetime difference we described earlier. These data confirmed not only that adhesion per se is essential for the generation of tension gradients, but also that adhesion strength, which scales with substrate stiffness[92], is also critical.

We then examined membrane tension gradients in non-adherent migrating cells, which typically migrate through bleb expansion. We confined HeLa cells between a cell-repelling glass coverslip and a PDMS blocked spaced by three microns, to induce leader-bleb migration. Cells display a gradient of cortical actin that directs persistent blebbing at the leading edge of the cell[93–95]. Long-range Flipper-TR lifetime gradients are absent in leader bleb migrating cells, over distances of 10–30 μm (Fig. 6a, d). A short-range (1–2 μm) gradient associated with the top-bottom difference are present (Supplementary Fig. 8a,b), analogous to the results in GUVs on PLL-g-PEG-coated glass. This shows that adhesion is necessary to create a tension gradient, suggesting that tension gradients are not essential for all forms of migration.

Clathrin plaques, distinct from transient clathrin-coated pits, are adhesive structures that rely on αvβ5 integrin-ECM interactions and depend on substrate rigidity, particularly on HeLa cells[96]. To test the role of clathrin plaques in the tension gradients, we patterned HeLa cells transiently expressing clathrin light chain fused with GFP and quantified the localization of the stable clathrin population using TIRF microscopy. The stable clathrin population, that we identify as plaques, contained only clathrin structures with a lifetime longer than 5 min. Clathrin plaques colocalized with low-lifetime zones, appearing at the inner edge and at the edges of the cross pattern (Fig. 6e). Treatment with cilengitide, which inhibits αvβ5-integrin-mediated

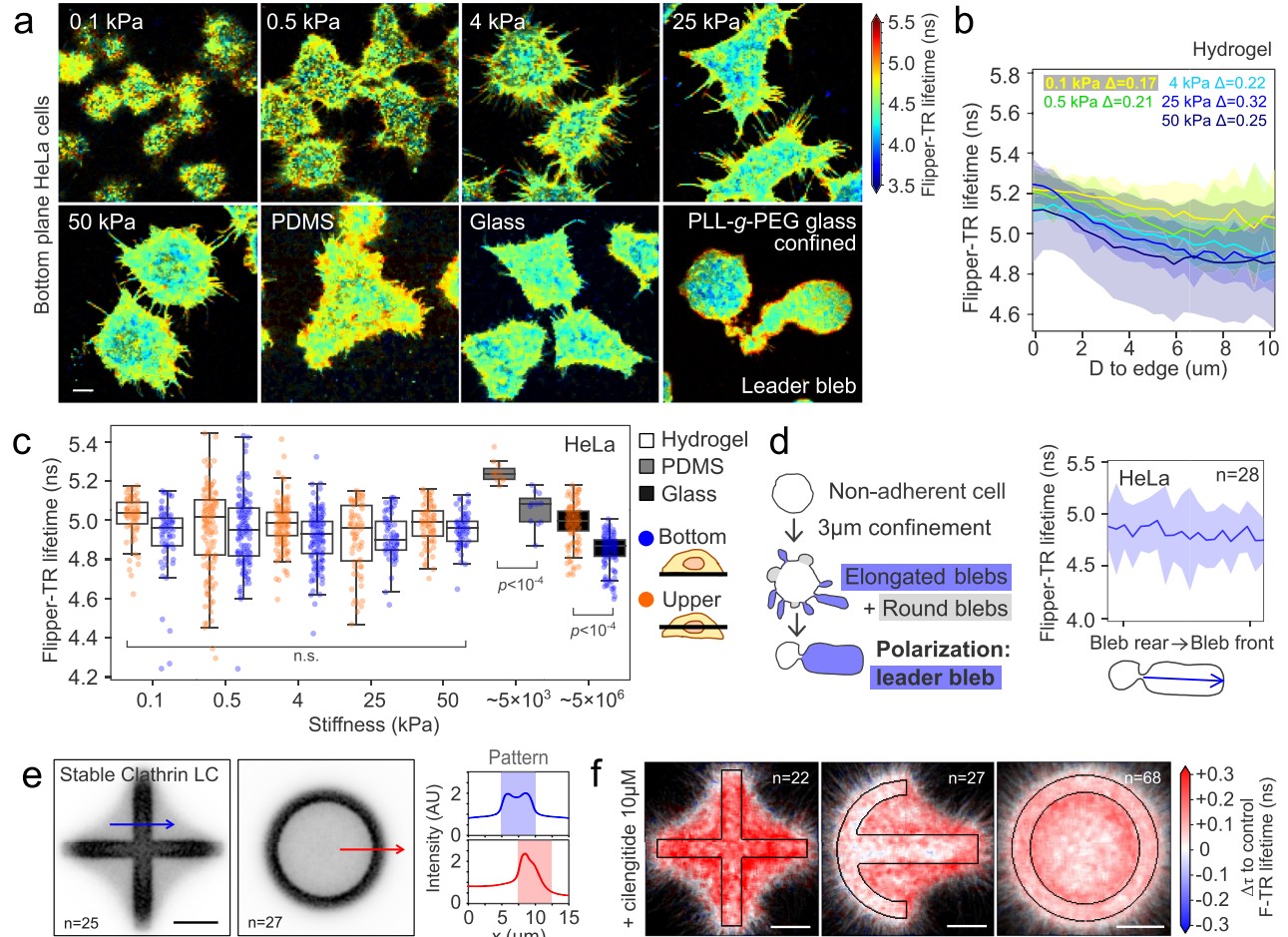

**Fig. 6 | Substrate rigidity and adhesiveness modulate membrane tension.**
**a** Representative live confocal FLIM images of the bottom plane of HeLa cells stained with Flipper-TR on different fibronectin-coated substrates. Color represents Flipper-TR lifetime. **b** Average Flipper-TR lifetime (ns) at the bottom plane, as a function of D, distance from the cell edge, in HeLa cells plated in fibronectin-coated hydrogels of varying rigidity. Labels represent the difference between the first and last 5 data points shown in the graph (-1.5 μm). Error represent SD. **c** Average Flipper-TR lifetime (ns) at the top and bottom planes of HeLa cells stained with Flipper-TR and a function of substrate stiffness (kPa). Median, quartile distribution, and individual data points are shown. $N = 3$ per condition, $n = 254$ total. Statistical test: Welch's p value. Boxplots show the median (center line), 25th and 75th percentiles (bounds of the box), and whiskers extending to the most extreme data points within $1.5 \times$ the interquartile range from the box. **d** Left, diagram of confinement experiment showing cell morphology and bleb formation before polarization into a leader bleb. Right, average Flipper-TR lifetime (ns) profile from the bleb rear to the bleb front, at the bottom plane, from HeLa cells confined to 3 μm under non-adhesive conditions. Error represent SD. **e** Average clathrin light chain intensity and line scans (blue lines) for cross (top, $n = 25$) and ring (bottom, $n = 27$) patterned cells. Patterned area is shaded. **f** Difference in Flipper-TR lifetime of cilengitide-treated cells from control average for cross, crossbow, and ring-patterned cells. **a, e, f** Scale bar, 10 μm.

adhesions, resulted in an increase of Flipper-TR lifetime at the areas surrounding the patterns (Fig. 6f), simulating the effect of plating cells on low-rigidity substrates. Thus, reducing adhesion strength by releasing molecular bounds flattens membrane tension gradients.

Overall, these results show that adhesion is required for creating tension gradients in cell membranes and that strength of adhesion can control the shape and intensity of the gradients.

## Discussion

By using Flipper-TR as a mechano-responsive molecular probe, here we established the conditions necessary to sustain spatial tension gradients within the bottom plasma membrane of adherent cells. Also, by employing in vitro experiments with model membranes, we could establish the exact nature of the coupling between Flipper-TR lifetime and membrane tension, and decouple the effect of actin and substrate interaction. Therefore, this work validates the use of Flipper-TR as an indicator of membrane tension and shows the influence of lipid composition on the relation between tension and lipid packing. We describe how branched actin networks and membrane-to-cortex

attachment are essential for tension buildup and tension propagation, respectively.

The differences between the upper and lower values of Flipper-TR lifetime gradients are -0.20 ns. From the calibration between flipper-TR lifetime and effective tension in HeLa[28,97], we can evaluate $\Delta\sigma_{eff}$ to be ~0.25 mN/m. Therefore, we estimate the tension to drop to be ~30–40% from the resting $\sigma_{eff}$ of 0.1–0.6 mN/m found in all cell types studied, which is far from being a negligible tension gradient. Within the same range, an increase of 30% higher at the leading edge compared to the trailing edge was previously measured in fish keratocytes by other means[12].

Some studies proposed that membrane tension propagates very fast, therefore homogenizing tension values[31], while others have proposed that tension gradients can be maintained in polarized cells[12,98]. Our work reconciles both ideas by proposing that tension propagation depends on the degree of membrane-substrate interaction. Studies that proposed fast diffusion were performed on cells in suspension[31], or focused on non-adhered membranes[9,17], while gradients were sustained only in adherent cells[12,18]. Therefore, contrary to what has been

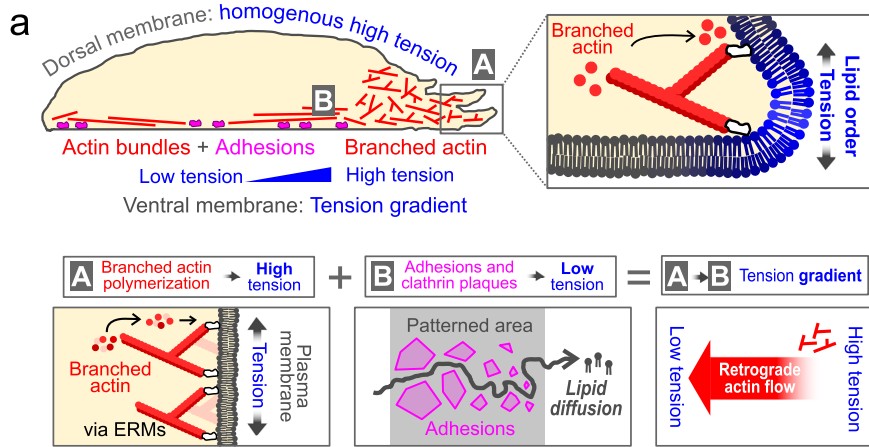

**Fig. 7 | Conceptual model. a** Conceptual model of membrane tension gradient maintenance. The diagram shows tension variation across the cell, highlighting high homogeneous tension on the upper side and tension gradients on the adhered side. Key factors include branched actin filaments (A, red) increasing tension, cellular adhesions and clathrin plaques (B, magenta) decreasing tension. The combination of these two create, creating tension gradients with actin flowing from high to low tension areas.

suggested[34], the existence of tension gradients does not depend on the cell type, but on the migration mode[95]. At the bottom membrane, attached to the coverslip, we find strong inhomogeneities, which disappear at the apical membrane not attached to the substrate (Fig. 7, top panel). This allows us to propose a mechanism by which membrane tension gradients can be sustained in adherent cells (Fig. 7). Dynamic extension of the membrane by actomyosin turnover at the cell edge locally increases the tension. The propagation of tension towards the cell center is hindered by adhesion to the substrate and between the membrane and the actin cortex. The constant pulling force at protrusion sites combined with the decrease of tension at adhesions set the shape of the tension gradient. In the following, we discuss the consistency of this model with previous literature.

We propose that cells must adhere to generate and sustain tension gradients. The remaining question is which are the molecular principles at play that limit lipid diffusion. While rapid lipid diffusion can relax tension gradients, tension could be relaxed by other mechanisms, such as membrane unfolding or the release of intracellular pressure[21,97]. Therefore, a reduction of lipid diffusion is not sufficient to create tension gradients, and thus assuming that hindered propagation of membrane flows is a signature of tension gradient may be fallacious[33]. Also, lipid diffusion depends on which scale it is measured. At the nanometer scale, no major difference in mobility is observed in lipid diffusion[99] between the upper and bottom membranes. At the submicron scale, lipid diffusion may be constrained by 'corrals' created by the mesh of cortical actin[100,101], by transient actin asters driven by Arp2/3[102,103], or by lipid nanodomains. The two last modes are very dynamic, so their contribution to the long-range flow barrier is less obvious. But it is really at the cell scale that lipid flows may counteract or create membrane tension gradients. This may explain discrepancies found in the literature on how lipid flows and membrane tension gradients are coupled, as the mobility of lipids is usually investigated at a molecular scale, while gradients are always evidenced at the cellular scale.

Regarding the role of actin, our data shows that branched actin both increases membrane tension at the leading edge and plays a role in its propagation. There is extensive evidence linking branched actin networks to membrane tension increase, often via ERM proteins[13,21,31,34]. What molecular mechanism mediates branched actin tension increase? On one hand, the branched actin nucleator Arp2/3 must bind to membrane-bound WAVE to add new filaments[104], linking actin polymerization and membrane binding. However, the specific properties of branched actin networks that promote tension increase might instead originate from their architecture, which confers them the ability to adapt density and pushing angle to the load to maintain polymerization[17,79,105].

Cell polarity is often studied in terms of membrane-associated signaling proteins, with less attention to the underlying physical state of the membrane. However, membrane organization can be highly sensitive to small tension changes, particularly near demixing transitions. The self-organized criticality at the plasma membrane effectively links membrane order to small perturbations like tension changes[59]. We show that branched actin increases tension and local membrane order, potentially promoting nanodomain formation. This could impact protein recruitment and signaling through changes in membrane partitioning, adding a mechanical layer of regulation to polarity.

Overall, our study demonstrates that membrane tension gradients in adherent cells arise from the interplay between actin dynamics, lipid diffusion, and cell-to-substrate adhesion. By visualizing tension gradients, we spatially link cytoskeletal forces to membrane mechanics and open new directions for studying how tension contributes to spatial control of signaling, polarity, and cell fate.

The limitations of our study stem from the use of fluorescence lifetime imaging microscopy (FLIM) on a scanning confocal setup. This technique is slow compared to the speed of actin dynamics, and especially to membrane diffusion (the acquisition time of single cells was ~30 s), which means that images of single cells are in fact averaged over times during which both actin and lipids move. Other FLIM techniques such as gated and modulated CCD image sensors might enable one to go faster and obtain images with higher time resolution[106]. While our results from MALDI-MSI on micropattern cells agree with lipid distributions expected from membrane tension gradients in vitro, it is important to note that MALDI-MSI requires fixation and quantifies lipids from all membranes, not just plasma membrane, factors that may interfere with the experimental interpretation. These factors may limit the use of MALDI-MSI to study lipid compositional gradients in live cells.

## Methods

### Mammalian cell lines

Human cervical adenocarcinoma cells HeLa-Kyoto (human female) stably expressing myosin IIA (Myh9)-GFP and LifeAct-mCherry or Myh9-GFP, or LifeAct-mCherry and the plasma membrane-targeting CAAX box fused to GFP, or TALEN-edited ActB fused with GFP (Cellectis, Paris, France), or HeLa, Cos7 (monkey male) and U2OS cells (human female) with no stable marker were cultured in DMEM

GlutaMAX medium (Gibco, #61965-026) supplemented with 10% FBS (Gibco, #P30-193306A) and Penicillin Streptomycin (Thermofisher, #15140-122) at 37 °C and 5% CO2. RPE1 cells (human female) were cultured in DMEM / F-12 (Gibco) supplemented with 10% FBS and 1% Pen-Strep at 37 °C in a 5% $CO_2$ incubator.

All cell lines were tested for mycoplasma contamination using Mycoplasmacheck PCR Detection (Eurofins, #50400400), and treated once a year with Plasmocin profilactic (InvivoGen, #ant-mpp) and treatment Mycoplasma Elimination Reagents (InvivoGen, #ant-mpt).

### Zebrafish keratocytes

Keratocyte cells were extracted from scales of zebrafish (adult males and females of *Danio Rerio AB*) while anesthetized with Ethyl 3-aminobenzoate methanesulfonate (Tricaine, SIGMA, E10521). 3-5 scales were extracted from each fish and placed on a glass-bottom dish containing Leibovitz's medium (L-15, ThermoFisher #21083027), supplemented with 10% fetal bovine serum and 1% penicillin/streptomycin antibiotic solution. The scales were subsequently sandwiched between the dish' glass and a coverslip and placed in an incubator at 28 °C. Keratocytes were allowed to spread out from the scales for 12 h. Subsequently, cells were washed twice with PBS and detached from the plates by treating them with trypsin-EDTA for 2–5 min. In order to remove the excess trypsin, cells were centrifuged in culture medium at 1000 rpm for 3 min. After discarding the supernatant, cells were resuspended in medium and plated in glass-bottom dishes for imaging. This protocol was carried out by trained personnel at the Zebrafish Core Facility of the medical center of the University of Geneva. Ethical approval was not required for this study because no animals were subjected to additional experimental procedures. Zebrafish scales were obtained exclusively as residual material from animals already used in ethically approved studies conducted at the University of Geneva under approval of CARE committee.

### DNA constructs and transfections

Cells were transfected with plasmid DNA using Lipofectamine 3000 reagent (Thermofisher #L3000001) transiently, according to manufacturer's protocol, and imaged from 24 to 72 h after transfection. The following expression vectors were used for plasmid DNA transfections: VAMP7-pHLuorin-pcDNA3 (gift from Thierry Galli, Inserm, France), eGFP-Clathrin Light Chain C1 (gift from P. De Camilli, Yale University, USA).

### Drug treatments and chemical probes

The following pharmacological inhibitors and chemical compounds were used: 10 µM of ezrin inhibitor NSC668394 (Sigma #341216), 20 µM of Rho Kinase inhibitor Y-27632 (Sigma #Y0503), 8 µM of actin filament stabilizer Jasplakinolide (Cayman, #CAY-11705-100), 5 µM of actin polymerization inhibitor Latrunculin B (Sigma, #428020), 1 µM of actin polymerization inhibitor Latrunculin A (Sigma, #L5163), 85 µM of Arp2/3 actin polymerization inhibitor CK-666 (Sigma #SML0006), 10 µM of $\alpha_V$ integrins inhibitor Cilengitide (Sigma #SML1594), and 10 µg/ml actin inhibitor Cytochalasin B for cell enucleation experiments (Sigma, #250255). Cells were imaged 30 min after treatment. CellMask Green plasma membrane stain (Sigma, #C37608) and Flipper-TR live cell fluorescent membrane tension probe (Lubio Science and Spirochrome, #SC020) were used according to manufacturer's specifications at 1:1000 from a 1 mM stock dilution in DMSO, to a final concentration of 1 µM. SiR-actin staining (Spirochrome #CY-SC001) was performed at 1 µM during 30 min with 10 µM verapamil following the manufacturer's instructions.

### Cell fixation and staining

Adherent cells were fixed for 15 min with 4% paraformaldehyde in PBS. Cells were then permeabilized and blocked with the blocking buffer [5% Bovine Serum Albumin and 0.1% Saponin in PBS] for 30 min. Cells

were washed three times with PBS before a 1 h incubation with Phalloidin-AlexaFluorPlus647 at 1:400 from a DMSO stock according to manufacturer's instructions (Thermofisher #A30107), Anti-Vinculin Vinculin Monoclonal Antibody (7F9)-Alexa Fluor488 at 1:25 dillution (eBioscience Thermofisher #53-9777-82) on 1% Gelatin in PBS. Cells were then washed three times with PBS, labeled by NucBlue Fixed Cell Stain (DAPI, Thermofisher #R37606) for 10 min, and washed again three times with PBS. Once stained, the cells were imaged directly on the microscope via the saved localization.

For lipid localization, we stained cells with fluorescently-labeled bacterial toxins that recognize different sphingolipid headgroups: Shiga toxin 1a-Cy3 (ShTxB1a, at 1:50) binds to Gb3[107], Cholera toxin-AlexaFluor488 (ChTxB, at 1:800) binds the ganglioside GM1[108], Equinatoxin II-AlexaFluor647 (EqTx, at 1:200) binds sphingomyelins[109] (all gifts from Giovanni D'Angelo, EPFL, Switzerland), and Filipin III at 25 µg/ml (Sigma, #F4767). The cells were fixed with 4% PFA, blocked in PBS containing 5% bovine serum albumin (BSA) without detergent, and incubated with fluorescently labeled B-subunit toxins for 1 h.

### Cell enucleation

The day before enucleation, U2OS and RPE1 cells detached with trypsin/EDTA (0.05% for 2 min, Gibco), and spread on a plastic polystyrene slide covered with bovine Fibronectin (FN, 10 µg/ml, Sigma-Aldrich #F4759) in their respective culture medium.

During enucleation, plastic slides seeded with cells were incubated at 37 °C for 15 min in the presence of 10 µg/ml Cytochalasin B (Sigma, #250255), then centrifuged at 15,000 g for 40 min at 37 °C in the presence of Cytochalasin. After centrifugation, to recover from the effects of Cytochalasin, the cells on plastic slides were washed with PBS, and transferred back to their respective drug-free culture media for 4 h at 37 °C before use.

### Migration time lapses of different mammalian cell lines

Mammalian cells were detached with trypsin/EDTA (0.05% for 2 min), then spread at a density not exceeding 10% confluence into 8-well chambered cover glass (Lab-Tek II) coated with bovine FN (10 µg/ml) in their respective culture medium, 1 to 3 h before the experiments.

Time-Lapse Phase-Contrast Microscopy Multisite microscopy of cells in 8-well chambered cover glass was performed in a humidified $CO_2$ chamber with an Axio Observer Inverted TIRF microscope (Zeiss, 3i) and a Prime BSI (Photometrics) using a 10X objective (Zeiss, 10X). SlideBook 6 × 64 software (version 6.0.17) was used for image acquisition. Cells were imaged using phase-contrast imaging, with a time-lapse of 5 min between frames for over 10 h.

### Confocal FLIM live imaging of Flipper-TR

During live FLIM acquisition, a corresponding phenol-red-free alternative of each culture medium was used. U2OS, HeLa and Cos7 cells were imaged in FluoroBrite DMEM (Gibco) during migration live experiments or Leibovitz's L-15 (Thermo Fisher Scientific, #21083027) during micropatterning experiments in incubation at 37 °C but without $CO_2$. Media was supplemented with 10 % FBS and 1 % Pen-Strep while RPE1 cells were imaged DMEM/F-12 without phenol red (Gibco) supplemented with 10 % FBS and 1 % Pen-Strep.

Generally, cells were labeled for at least 10 min with FBS-free medium containing 1 µM Flipper-TR, following the manufacturer's instructions. For supported lipid bilayers, the medium consisted on an aqueous buffer of 10 mM HEPES at pH 7.4. After labeling with Flipper-TR or Flipper-TR and Hoechst, cells were imaged directly in glass-bottom microwell dishes at 37 °C and 5% CO2. For migration assays, untreated and enucleated cells were detached with trypsin/EDTA (0.05% for 2 min), then spread at a density not exceeding 20% confluence into 35 mm glass-bottom microwell dishes (MatTek) coated with bovine FN (10 µg/ml) in their respective culture medium, 1 to 3 h before the experiments. After spreading, the untreated cells were

labeled with Flipper-TR (Spirochrome) in their respective acquisition medium and incubated at 37 °C for 10 min without probe washing. The Flipper-TR stock solution was composed by dilution of a 1 mM Flipper-TR in DMSO as previously described[28]. After spreading, the enucleated cells were labeled by 1:1000 dilution of a stock solution of Flipper-TR and labeled by NucBlue Fixed Cell Stain (DAPI, Thermofisher #R37606) in their respective acquisition medium and incubated at 37 °C for 10 min without probe washing.

The microscope was an inverted motorized microscope Eclipse Ti2 (Nikon) with a point scanning A1 confocal system from Nikon, equipped with a Plan Apo λ 100x oil objective with a NA 1.45 (Nikon, # MRD01905), a perfect focus system, a stage-top incubation chamber (Okolab) allowing long acquisitions. The imaging on hydrogels and comparing experiments was done instead with a water immersion 40x objective NA 1.15 (Nikon, # MRD77410) to allow for a higher working distance. The A1 confocal system was equipped with 4 excitation lasers at 405, 488, 561 and 640 nm. For fluorescence lifetime imaging (FLIM), we used a time-correlated photon counting system from PicoQuant integrated on Nikon Elements software. Excitation was performed using a pulsed 485 nm laser (PicoQuant #LDH-D- C-485) mounted on a laser-coupling unit, and a computer-controlled multichannel picosecond diode laser driver (PicoQuant, Sepia PDF 828) operating at 20 MHz. The emission signal was collected through a 600/50 nm bandpass filter using a gated PMA hybrid 40 detector and a multichannel time-correlated single photon counting system (PicoQuant, MultiHarp 150).

Single frames resulted from the integration of 2–10 frames depending on the sample and the confocal scanning area, to a total count between 10 and 50 photons per pixel. Frame rate during time lapse imaging in mammalian cells had to be set at 10 min to avoid phototoxicity and arresting actin dynamics.

### Confocal live imaging for fluctuation correlation spectroscopy

Regarding fluctuation correlation spectroscopy analysis, the microscope setup was the same as for FLIM. For both line scanning used for autocorrelation function (ACF) and paired-correlation function (pCF) analysis, pixel dwell time was set at 5.2 microseconds, line time at 960 microseconds, pixel size was set at 160 nm, giving a line length of 10 micrometers. We positioned the 10-micrometer confocal line scans crossing the pattern boundaries, thus across regions with a sharp decrease of the Flipper-TR fluorescence lifetime. For the ring pattern, the line scan sits at the cell edge in the region where retrograde flows are dissipating. The adhesive area is located on the outer side of the line scan. For the cross pattern, the line scan is located closer to the center of the cell, and the adhesive pattern is on the inner side of the line.

### Preparation of supported lipid bilayers

For supported lipid bilayer experiments, we used: 18:1 (Δ9-Cis) PC (DOPC) (Avanti #850375), 18:1 PS (DOPS) (Avanti #840035), porcine brain SM (Avanti #860062), plant cholesterol (Avanti #700100) at the specified mol% ratios. Some lipid mixtures included 0.02 mol% Atto 647 1,2-Dioleoyl-sn-glycero-3-phosphoethanolamine (DOPE) (Sigma #67335). Phospholipids or cholesterol stocks were dissolved in chloroform at concentrations ranging from 1–10 mg/ml and stored at −80 °C in Argon atmosphere. Lipid mixes were calculated for a final mass of 0.25 mg. The chloroform in the vials was evaporated under an Argon flow in a fume hood and lipid films were formed. The vials were then stored in a vacuum oven at room temperature overnight to remove any residual chloroform.

To prepare liposomes, the vials were brought to room temperature and rehydrated by adding aqueous buffer HEPES 10 mM at pH 7.4 to a final concentration of 0.5 mg/ml of lipids and mixing. The liposome mixtures were placed on a parafilm surface. 50-μm silica particles (Sigma #904384-2 G) were first diluted 3x with distilled water, washed

three times, and deposited on the lipid droplets on parafilm. The liposome-beads aqueous mixtures were then stored in a vacuum oven at room temperature overnight to remove any residual chloroform. To prepare the spreading bilayer assay, culture-grade 35-mm glass-bottom dishes (Mattek #P35G-1.5-14-C) were plasma cleaned (Harrick Plasma #PDC-32G) for 2 min at high power. Immediately after, dried liposome-bead mixtures were scrapped from the parafilm surface, deposed on the plasma-cleaned surface, and rehydrated with 1 ml of 10 mM HEPES pH7.4 buffer containing 1 μM Flipper-TR (Lubio Science and Spirochrome, #SC020).

### TIRF-FRAP experiments on patterned cells

All total internal reflection fluorescence microscopy (TIRF) movies were recorded on an Olympus IX83 widefield microscope equipped with a 150×/NA1.45 objective and an ImageEM X2 EM-CCD camera (Hamamatsu) under the control of the VisiView software (Visitron Systems). The 488 nm and 561 nm laser lines were used for illumination of GFP- and mCherry-tagged proteins. Excitation and emission were filtered using a TRF89902 405/488/561/647 nm quad-band filter set (Chroma). Laser angles were controlled by iLas2 (Roper Scientific).

Fluorescence recovery after photobleaching (FRAP) experiments were performed using a custom-built set-up that focuses a 488-nm laser beam at the sample plane, on the Olympus IX81 widefield microscope described above. The diameter of the bleach spot or width of the bleach line was approximately 0.5 μm.

### Chemical passivation and micropatterning

Glass surfaces passivated with Polyethyleneglycol (PEG) were prepared for protein micropatterning. First, glass bottom dishes (Mattek) were activated by exposing them to air plasma (Harrick Plasma, PDC-32G) for 3 min. Subsequently, the dishes were treated with a 0.1 mg/ml poly-lysine (PLL) (Sigma) solution for 30 min and washed with 10 mM HEPES buffer (pH 8.4). By using this same buffer, a solution of 50 mg/ml polyethylene glycol (PEG) (molecular weight 5,000) linked to a succinimidyl valerate group (SVA, Laysan Bio) was prepared and applied to passivate the PLL-coated surface for 1.5 h. Finally, dishes were washed with PBS 3 times.

Micropatterns were generated by using the system PRIMO (Alvéole), mounted on an inverted microscope Nikon Eclipse Ti-2. In the presence of a photo-initiator compound (PLPP, Alvéole) and DMD-generated patterns of UV light (375 nm), PEG is degraded. After illumination (1250 mJ/mm) through a 20x objective, PLL is exposed. After rinsing with PBS, fibronectin (Calbiochem) was incubated at 50 μg/ml at room temperature for 5 min to coat the PEG-free motifs with the cell-adhesive protein. The excess of fibronectin was washed out with PBS. PBS was finally replaced by medium, and a suspension of cells was added at densities of roughly $10^5$ cells per $cm^2$. Samples were kept in an incubator at 37 °C and 5% CO2. After 10–30 min, non-adhered cells were washed out.

### MALDI-MSI sample preparation

Cells were directly seeded on micropatterns with cross shape in complete media. After aspiration of media, cells were washed twice with PBS, followed by fixation in 0.25% glutaraldehyde for 15 min. For MALDI-MSI analyses, 150 μL of 2,5-dihydroxybenzoic acid (DHB), (30 mg/mL in 50:50 acetonitrile/water/0.1% TFA), were deposited on the surface of the samples using the automatic SMALDIPrep (TransMIT GmbH, Giessen, Germany).

### Cell culture on hydrogels

Commercially-available EasyCoat hydrogels (Cell Guidance Systems, # SV3510-EC-0.5-EA and # SV3510-EC-4-EA) were used to control substrate rigidity. The hydrogels were pre-coated with fibronectin to facilitate cell adhesion. Briefly, hydrogels of specified rigidity (0.1, 0.5, 4.0, 25, and 50 kPa) were equilibrated in PBS-fibronectin

solution (1 μg/mL, Sigma #F1141) for 1 h to ensure uniform coating. Excess fibronectin was washed off with PBS before seeding the cells onto the hydrogels. Cells were cultured on these substrates under standard conditions, allowing us to investigate the effects of substrate rigidity on membrane tension and cell behavior.

## Cell confinement

Cell confinement was achieved using a PDMS micropillar array mounted on a coverslip mounted on a PDMS device (named 'suction cup') trigged by a vacuum pump and pressure controller, following established methods[93,110]. Briefly, a SU8 photolithography mold was prepared with micropillars. This mold is used to create a PDMS confinement chamber of 3 micrometers in height, plasma-bonded to a 12-mm glass coverslip. The microfabricated confiner coverslips were treated with plasma for 1 min, and incubated with 0.5 mg/mL pLL-*g*-PEG (SuSoS, PLL(20)-g[3.5]-PEG(2)) in 10 mM pH 7.4 HEPES buffer for 1 h at room temperature. Cells were trypsinized and seeded in a glass-bottom dish coated with PLL-g-PEG and the PDMS device was placed on top, creating a confined environment between the glass and the PDMS coverslip. This setup allowed us to trigger leader-bleb migration by confining the cells to a 3-micrometer height, promoting a non-adhesive environment conducive to the desired migration behavior.

## Statistical analysis of experimental data

Statistical analyses for all experiments were performed in Python 3 (NumPy and SciPy libraries) or Microsoft Excel. Statistical data are presented as median or mean ± standard deviation or interquartile range. For each panel, sample size (*n*), experiment count (*N*), statistical tests used, and *P* values are specified in the figure and/or legends. Plots were made using the SciPy and matplotlib Python libraries.

## Basic image processing

Basic image analysis and format conversion was performed on ImageJ/Fiji software. Images were imported from NIS Elements (.nd2 format) using the BioFormats plugin.

## Migration parameters and cell morphology of mammalian cell lines

Individual cells were tracked semi-automatically by random selection of cells from video images and manual tracking of migration pathways using the Manual Tracking function in ImageJ. At least 50 cells were tracked for each cell type ($n \geq 50$). The cell migration tracking data were analyzed as described previously[111]. The directionality ratios are calculated by taking the ratio of the displacement of the cell and the length of the actual path it took.

*Cell aspect ratio* was calculated from binary masks using Image/Fiji by fitting ellipses to the cell shape.

*Protrusion area* in control patterned cells was segmented manually from intensity images.

## General considerations about the different Flipper-TR lifetime estimates

Flipper-TR is a mechano-responsive molecular probe that changes its fluorescence properties in response to changes in membrane packing, making it an invaluable tool for studying the mechanical properties of the plasma membrane. Upon insertion to the membrane, Flipper-TR's fluorescence lifetime changes with the tension in the lipid bilayer. Generally, the fluorescence lifetime is the average time the molecule spends in the excited state before emitting a photon and returning to the ground state. To calculate the fluorescence lifetime of Flipper-TR, we used a time-correlated single photon counting (TCSPC) device as described above. This technique measures the time between the excitation pulse and the subsequent photon emission, producing as an output the histogram of photon arrival times. The distribution of these arrival times can be analyzed to determine the fluorescence lifetime.

Due to the photochemistry of the molecule, Flipper-TR photon arrival time distribution follows a bi-exponential decay $I(t)$, in the form: $I(t) = A_1 e^{-t/\tau_1} + A_2 e^{-t/\tau_2}$, where $A_i$ and $\tau_i$ represent the amplitude and the decay of a given exponential $i$, respectively. Previous studies have employed different methods to estimate the Flipper-TR lifetime. One common approach is to fit the fluorescence decay curve to a biexponential decay model, which accounts for the presence of two distinct exponential decay components[28]. From this model, researchers can extract either the larger exponent (i.e., $\tau_1$) or calculate an intensity-weighted average of the two exponents, $\bar{\tau} = \sum_i \tau_i \alpha_i$[97]. These methods can provide detailed insights into the Flipper-TR lifetime, including a $\chi^2$ used to judge the goodness, a greater precision, and the ability to decompose complex lifetime distributions, but they require more computational resources and can be more sensitive to noise and initial parameter estimates. Moreover, even at high photon counts, the ability to determine the precise values of $\alpha_i$ and $\tau_i$ by a biexponential fit can be hindered by parameter correlation[112]. To avoid problems involved with fitting, other works have used phasor plots have been used to measure changes in Flipper-TR lifetime[113,114]. Phasor plots offer a graphical representation of the fluorescence decay characteristics, simplifying the analysis of complex decay patterns. By mapping each pixel to specific G/S coordinates on the phasor plot, one can visually assess changes in membrane tension and identify different lifetime components without assuming any feature of the fluorescence decay, such as the number of exponents. This is particularly useful when using complex samples or Flipper variants where the photochemical properties have not been fully characterized.

For this work, we have instead relied on the barycenter of the lifetime distribution (i.e., mean photon arrival time or "fast lifetime" calculation) to estimate fluorescence lifetime. As the instrumentation induces a delay in the photon arrival, the average lifetime equals time span from the barycenter of the instrument response function (IRF) to the barycenter of the decay in a pixel-wise manner, as defined by our TCSPC manufacturer (PicoQuant). This method is simple, straightforward, and computationally efficient, especially reliable at low photon counts. The calculation yields a single lifetime exponent, that we refer to as "average Flipper-TR lifetime" $\bar{\tau}$, numerically equivalent to the intensity-weighted average of the two exponents coming from a fit. The need for a robust lifetime estimator at low photon counts was very important for this work as we were highly limited by sample illumination. Protrusion dynamics are particularly sensitive to phototoxicity and even relatively low illumination can arrest actin polymerization, so the photon counts were generally too low to be able to fit in a robust manner.

## Flipper-TR lifetime estimation

*Processing and plotting of FLIM images.* FLIM images acquired on NIS software consisted of a channel containing average lifetime values and another with photon counts per pixel. The images were exported as.tif and processed on a custom-made Python script. Lifetime images were applied to a 2D median filter with a kernel size of 3 pixels (corresponding to 0.75 μm). In the exported lifetime images, the rainbow color scale in each pixel represents the values of the decay times. On top of the rainbow color scale, a dark mask is applied to represent photon counts (i.e., black for no counts). This allows for a proper visual representation of the data because it masks the background where lifetime values are just due to noise.

**Flipper-TR lifetime region-of-interest averages.** When averaging pixels, the lifetimes of each pixel $\tau_i$ were weighted by the photon count $n_i$, so that $\bar{\tau} = [\sum_{i=1}^{n}(n_i \tau_i)]/N$. Different regions corresponded to specific criteria, explained as follows. In keratocytes, front/rear regions were manually segmented to cover approximately one-fourth of the

cell each. In migrating mammalian cells, front/rear regions were defined as 60° angular sections extending 0-10 μm from the edge using a custom python script. Front and rear polarity angles were manually determined based on cell morphology, not lifetime values. For supported lipid bilayers, only the 100 μm closest to the advancing edge in the field of view was considered, using a custom Python script. Front and rear regions were defined as the 20% closest and most distant areas from the advancing edge, respectively.

**Flipper-TR lifetime distribution and ROIs in patterned cells.** Generally, FLIM images were filtered and values under a threshold photon count of 20 photons were defined as NaN and not considered for analysis. For overall distribution, i.e., the lifetime spatial probability map, cells were aligned according to the patterned regions[115], and a pixel-wise average was calculated using a custom python script. For cells adhered on micropatterns, different regions were defined to report spatial lifetime gradients: bottom/upper planes; cell body/ protrusions; adhesive/non-adhesive; high/low. For the bottom plane there was no further segmentation, but for the upper plane (3 microns above the glass surface) a manual selection was used to segment the plasma membrane. Cell body and protrusions were manually segmented. Adhesive and non-adhesive regions were defined based on the fluorescence from micropatterns. Finally, to calculate the distance map 2D histogram, distances to the selected boundaries were defined per pixel using binary masks and the 'distance map' function in ImageJ/ Fiji. Then, pixels were ordered and averaged to form a 2D histogram. To define the high and low regions we first defined the pixels were 99% of control cells were present. From this, we defined a 'high' region, containing the highest two lifetime deciles, and a 'low' region, containing the lowest 2 lifetime deciles of the control lifetime spatial distribution. These two regions of interest were applied to other conditions to represent lifetime gradients.

*Flipper-TR lifetime spatial decays* in mammalian cell lines were calculated using a custom python script based on distances to the cell edge and position in the micropattern. For radial/linear gradients, a 5 micron-wide region was defined and the average lifetime calculated per micron.

*Flipper-TR lifetime as a function of edge position* in migrating cells was calculated using a cell mask based on photon counts and, from a mask, a distance map to the edge. Then, pixels were ordered and averaged to form a 2D histogram.

**Flipper-TR lifetime as a function of edge velocity.** The edge velocity was calculated using a custom Python script. For mammalian cells, the edge displacement was calculated by segmenting the cell contours at each timepoints and calculating the differences between the distance maps. For supported lipid bilayers, the images were manually cropped into rectangular regions. The velocity at each timepoint (μm/min) was defined as the net increase of area (μm²), divided by the frame rate (min) and the crop width (μm).

**Flipper-TR lifetime in protruding/retracting cell regions.** The condition to classify edges into protruding/retracting was a velocity higher/lower than 0.2 μm/min. From this, angular sections extending 0–10 μm from the edge were defined.

**Flipper-TR lifetime differences to control distributions.** For some drug treatments, instead of the absolute Flipper-TR distributions, the difference to the control was instead plotted. First, the average control distribution was calculated on a pixel-wise basis. Then, these control values were subtracted to the drug-treated images on a cell-by-cell basis and the average differences plotted.

*Flipper-TR lifetime in hydrogels and non-adhesive migration* was instead calculated using the SymPhoTime 64 software (PicoQuant) to fit fluorescence decay data (from manual regions of interest) to a dual exponential model (n-exponential reconvolution) after deconvolution for the instrument response function calculated by the software. Lifetime was then expressed in difference in nanoseconds from the reference value, which is the lifetime at the upper plane of cells adhered to glass (3 microns above glass surface).

### Particle image velocimetry of actin flows

*PIV analysis.* Movies were acquired at a 2-frame-per-second rate and segmented using Ilastik[72] to define cell boundaries. Pixels outside cell boundaries were not considered for the PIV calculation. PIV vectors were calculated on a custom macro based on the 'Iterative PIV' ImageJ/ Fiji plugin[116], using approximately an XY grid of 1.25 μm and a search area of 1.5 μm. Importantly, even though we used the 'Iterative PIV' plugin, the PIV was calculated on a single window basis, so there is no a priori correlation between neighbor vectors.

### Quantification of membrane events

*Stable clathrin coverage* was calculated from a minimum time projection of TIRF movies over 5 min at a 1 frame/second rate. This projection was then binarized to display the sites where the clathrin signal was stable over 5 min on a cell-by-cell basis.

*Dynamic clathrin coverage* was calculated in an analogous manner. First, we obtained a binary mask resulting from a maximum projection of the clathrin signal. Then we subtracted to this mask the stable clathrin sites. This yields a mask containing the sites where the clathrin signal was dynamic at some point over 5 min on a cell-by-cell basis, excluding the clathrin plaques where the signal was stable.

*VAMP7 exocytosis events* were manually selected from 5-min TIRF movies of VAMP7 transfected and patterned cells.

Finally, the probability density distributions were plotted as a function of the distance to the center of the pattern and normalized to integrate to 1.

### Actin-based segmentation of cell regions

To determine the effect of actin organization on Flipper-TR lifespan. Phalloidin-A647 images of each cell were segmented into three regions; Lamellipodium, Lamella, and Cell body regions. Binary segmentations were generated using the pixel classification process in Ilastik[72]. Segmentation masks were then applied to the Flipper-TR lifetime image acquired before fixation, to calculate the average Flipper-TR lifetime in each type of region for each cell.

### Fluctuation correlation spectroscopy analysis

To study lipid diffusion in our system, we employed fluorescence fluctuation spectroscopy (FFS). This involved first construction of single-channel FFS line scans acquired across the bottom membrane of patterned cells stained with the plasma membrane dye CellMask into kymographs (x-dimension represents distance (i.e., 64 pixels), y-dimension represents time (i.e., 100,000 lines)) and application of a detrending algorithm to remove slow timescale artefacts (e.g., photobleaching). Then application of the autocorrelation function (ACF) on a pixel-by-pixel basis ($\delta r = 0$) to three spatially distinct sections across the kymograph (i.e., pixels 1-21, 22-42, and 43-63), which alongside fitting of the resulting ACFs to a 2-component diffusion model, enabled recovery of the average number of moving molecules ($G_0$) as well as their local diffusion coefficient ($D_0$) to be mapped across the non-adhesive versus adhesive coating and the interface in between (Fig. 5E, right panel). And finally, application of the pair correlation function (pCF) on a pixel-by-pixel basis ($\delta r = 5$) across the entire kymograph (i.e., pixels 1-59 from left to right and pixels 5-64 from right to left), which alongside fitting of the resulting pCFs to a Gaussian function, enabled recovery of the molecules direction dependent arrival time ($\tau$) as well as their transport efficiency (i.e., $G_\tau / G_0$) (Fig. 5E, left panel). Published codes are also available at https:// github.com/ehinde/Pair-correlation-microscopy.git repository.

## Analysis of TIRF-FRAP experiments on patterned cells

FRAP experiments performed on MyrPalm-GFP patterned HeLa cells were analyzed as previously described[117]. Mean fluorescence values were measured from regions of interest representing the background, the cell, and the membrane. A custom-written Python script was used to subtract background fluorescence, correct for photobleaching, and normalize the values between 0 (corrected fluorescence immediately after photobleaching) and 1 (mean corrected fluorescence of 5 s before photobleaching). The recovery curves of individual experiments were aligned to bleach time and averaged. The average was fitted to a single exponential equation from which the mobile fraction and recovery half-time were calculated.

To calculate the velocity of lipid flows on line-FRAP experiments, fluorescence intensity was integrated over 11 microns (100 pixels) parallel to the bleaching line, and the minima were tracked for 12 frames (0.5 seconds per frame) using a custom-made Python script.

## MALDI-MSI data analysis

MSI experiments were performed using AP-SMALDI5 AF systems that couple a Q Exactive orbital trapping mass spectrometer (Thermo Fisher Scientific, Bremen, Germany) with an atmospheric-pressure scanning-microprobe MALDI imaging source (AP-SMALDI, TransMIT GmbH, Giessen, Germany). The MALDI laser focus was optimized manually using the source cameras aiming at a diameter of the focused beam of 7 μm. For each pixel, the spectrum was accumulated from 50 laser shots at 100 Hz. MS parameters in the Tune software (Thermo Fisher Scientific) were set to the spray voltage of 4 kV, S-Lens 100 eV, and capillary temperature of 250 °C. The step size of the sample stage was set to 5 μm. Positive ion mode measurements were performed in full scan mode in the mass range $m/z$ 400-1600 with a resolving power set to R = 240000 at $m/z$ = 200. Mass spectra were internally calibrated using the lock mass feature of the instrument.

The output of the MALDI-MSI was processed using a custom-made Python script. The output consists of a multi-dimensional.tif with 389 channels representing the peak intensity at each $m/z$ bin and XY spatial coordinates at 0.5 microns per pixel. First, an integration of all counts allowed to identify pattern positions with cells and to align patterns with one another. Then, the relative intensity of peaks from specific lipid species was calculated over the total intensity of only the known peaks (112/389). For lipid species present in more than one peak, the relative abundances were added together. This yields a lipid fraction per pixel, per cell, of each lipid species. Lipid fractions were also normalized to analyze their spatial heterogeneity. To do so, lipid fractions from a given lipid species from all cells, all pixels were pooled together, and their mean and standard deviation were calculated. Lipid fractions were then subtracted from the mean and divided by the standard deviation.

## Ethical statement

This research complies with all relevant ethical regulations.

## Reporting summary

Further information on research design is available in the Nature Portfolio Reporting Summary linked to this article.

## Data availability

The microscopy image data and associated datasets generated in this study have been deposited in the Figshare repository under accession code: doi.org/10.6084/m9.figshare.30174265. The processed datasets and source data underlying the figures are available in the same repository. Any additional information required to reanalyze the data reported in this paper is available from the lead contacts upon request.

## Code availability

All code files have been deposited in the Figshare repository: https://doi.org/10.6084/m9.figshare.30174265.

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

## Acknowledgements

We thank Martin Lenz for discussions on membrane flows, Julie Miesch and Anne-Laure Boinet for TIRFM training, and Chloé Roffay and Vincent Mercier for FLIM training. Thanks to Rafael Caetano, Frédéric Humbert, and Roux lab members for technical assistance and feedback. Special thanks to Mireia Andreu Carbó, Guillaume Pernollet, Vincent Mercier, and the ACCESS platform for micropatterning training. We thank Henry De Belly, Matthieu Piel, Mathieu Dedenon, Chloé Roffay, Larisa Venkova and Jérémie Francfort for critical reading. The authors acknowledge the following funding: #CRSII5_189996, Swiss National Science Foundation (SNSF), AR. #310030_200793, Swiss National Science Foundation (SNSF), A.R. #951324-R2-TENSION, European Research Council (ERC) Synergy Grant, A.R. LT000762/2020-L, Human Frontiers of Science Program (HFSP), A.M. ALTF 989-2022, European Molecular Biology Organization (EMBO) Long-Term Fellowship, J.E. LT-000793/2018-C, Human Frontiers of Science Program (HFSP), P.G. PID2022-140687NB-I00, MICIU/AEI/10.13039/501100011033 and FEDER, UE, A.C. PID2022-140687NB-I00 funded by MCIN/AEI /10.13039/501100011033/ and by FEDER, UE, A.C. CNS2024-154624, MICIU/AEI/10.13039/501100011033, A.C. IT1625-22, Basque Government, A.C. #31003A_182473, Swiss National Science Foundation (SNSF), C.A. #TMSGI3_211433, Swiss National Science Foundation (SNSF), C.A. DIP funding, Canton of Geneva, CA.

## Author contributions

J.M.G.A.: Conceptualization, Methodology, Software, Formal Analysis, Investigation, Data Curation, Writing – Original Draft, Writing – Review & Editing, Visualization, Project Administration. A.M.: Conceptualization, Investigation, Software, Formal Analysis, Visualization. J.S.: Formal Analysis. P.G.: Conceptualization, Investigation, Writing – Review & Editing. C.T.: Conceptualization, Investigation, Writing – Review & Editing. L.H., L.C., J.E.: Investigation. G.'A.: Supervision, Funding Acquisition. A.C.: Conceptualization & Investigation. E.H.: Software, Formal Analysis, Supervision, Funding Acquisition. C.A.: Conceptualization, Supervision, Project Administration, Funding Acquisition. A.R.: Conceptualization, Methodology, Writing – Original Draft, Writing – Review & Editing, Supervision, Project Administration, Funding Acquisition.

## Competing interests

The authors (J.M.G.A., A.M., J.E., C.A., A.R.) declare the following competing interests: the University of Geneva has licensed Flipper-TR probes to Spirochrome for commercialization. The remaining authors have no competing interests to declare.
