## [Transparent Peer Review file · Nature Communications]

Adherent Cells Sustain Membrane Tension Gradients Independently of Migration

Corresponding Author: Professor Aurélien Roux

Version 0:

Reviewer comments:

Reviewer #3

(Remarks to the Author)

The authors have satisfactorily addressed my previous comments.

(Remarks on code availability)

Reviewer #4

(Remarks to the Author)

This reviewer has been requested to assess whether the concerns by Reviewer #1 have been addressed by the authors.

I agree with R1 that this is a beautifully executed study, combining FLIM of Flipper sensors with other orthogonal techniques to provide a mechanistic insight into how actin scaffolds and substrate adhesion can lead to membrane tension gradients, even in the absence of cellular migration. I believe most of R1 concerns have been addressed, and the authors provide a convincing case for publishing this work in Nature Communications.

Major comments:

- When discussing the bleb data (Fig. 6d) R1 mentions the difference in lifetime between the inside/membrane is ~1.5ns (confirmed in Extended Fig. 8), which still looks higher than for GUVs (Extended Fig. 3). How could this be explained, assuming the GUV compositions used are representative of the plasma membrane composition? Also, it appears to me that lifetime inside the non-adherent cell is lower than for the bleb's inside. Is this something that the authors see across all samples? If so, how would they justify it?
- In Extended Fig.3d, the authors show a lack of lifetime gradient in DOPC/DOPS membranes and hypothesise this is due to the fluid state of the membrane. However, this composition also contains the charged lipid DOPS. Repeating the experiment with pure DOPC GUVs could help to clarify this. Further, I am surprised by the GUV arrangement. DOPC/DOPS is clearly non-spherical, while bSM-containing membranes seem to aggregate. Could the author please comment on this?
- Fig. 6, HeLa cells seeded on 0.1kPa hydrogels appear not to display a tension gradient. However, I wonder whether this observation could result from an artifact related to the smaller cell size and/or more rounded cell morphology (e.g. more membrane planes are potentially contributing to the FLIM acquisition). Could the authors, try to explore this, e.g. by extracting a decay constant for each of the gradients, and plotting against the cell area?
- Line 134, please clarify which compositions are close to the demixing transition. Provide references and/or experimental data (e.g. DSC) for the melting temperature of these lipid mixtures.
- Line 227, the authors indicate that differences in actin cytoskeleton morphology led to different tension values, as evidenced by different Flipper lifetimes. However, as pointed out by R1, molecular rotors (sensitive to lipid packing) also show a higher lifetime in GUVs in the presence of actin. Because Flipper must also report on lipid packing (given its distinct lifetimes in membranes of different composition), how can the authors rule out the possibility of different lipid packing caused

by the attached acting vs a tension gradient in the membrane?

- For the MALDI-ToF experiments, Gb3 (a bulky lipid) seems to accumulate at the edges/protrusions, where tension is also higher. The composition of the plasma membrane is believed to be close to the critical points – hence allowing the formation of transient nanodomains. Hence, why Gb3 accumulates at the edges while DOPE-A647 is depleted towards the edge of SLBs corresponding to 1:1:2 and 1:1: DOPC:bSM:Chol composition?

Minor comments:

- Fig. 6d (bleb) – Why does the profile remain flat? From the FLIM image, I would expect an increase in Flipper lifetime at the front's edge (similar to Extended Fig. 8a)
- Replace “demixing compositions” (e.g. Line 150) for compositions far away from the transition temperature.
- Out of curiosity, why did the authors decide to form SLBs from coated-beads? Why not directly from SUVs, or GUVs? In the later case, the authors would have been able to observe the spread of microscopic Ld and Lo domains.

(Remarks on code availability)

Version 1:

Reviewer comments:

Reviewer #4

(Remarks to the Author)

The authors have successfully addressed my comments, and I fully support the publication of this outstanding piece of work in Nature Communications.

(Remarks on code availability)

N/A

**“Adherent Cells Sustain Membrane Tension Gradients Independently of Migration”
Nature Communications NCOMMS-25-52487A**

POINT-BY-POINT RESPONSE TO REVIEWER’S COMMENTS

Reviewer #3 (Remarks to the Author):

The authors have satisfactorily addressed my previous comments.

We thank the positive feedback of the reviewer.

Reviewer #4 (Remarks to the Author):

This reviewer has been requested to assess whether the concerns by Reviewer #1 have been addressed by the authors.

I agree with R1 that this is a beautifully executed study, combining FLIM of Flipper sensors with other orthogonal techniques to provide a mechanistic insight into how actin scaffolds and substrate adhesion can lead to membrane tension gradients, even in the absence of cellular migration. I believe most of R1 concerns have been addressed, and the authors provide a convincing case for publishing this work in Nature Communications.

We thank the positive feedback of the reviewer on these aspects.

Major comments:

- When discussing the bleb data (Fig. 6d) R1 mentions the difference in lifetime between the inside/membrane is ~1.5ns (confirmed in Extended Fig. 8), which still looks higher than for GUVs (Extended Fig. 3). How could this be explained, assuming the GUV compositions used are representative of the plasma membrane composition?

Reviewer 1 comments on the 1.5ns difference referred to the noise of the first version of the paper, before the revision works; Verbatim: “the image shows a huge variation in lifetime (+/-1.5 ns, compared to the typical changes of 0.1-0.2 ns with tension) and to my eye ‘the inside’ of the cell is more blue than the outside, which is contrary to the author’s conclusions”. Reviewer 1 also pointed that the plasma membrane at the edge of the contact patch showed lower lifetime (see below, left is previous image on the first version, and right is the revised panel). To answer Reviewer’s 1 questions, we took more resolved images on bleb-migrating cells, which is challenging due to time resolution imposed by FLIM. These data below show that increasing photon counts reduces dramatically the variation.

Second, the difference of lifetime between the cell/bleb edges to the inner part increased. This is consistent with the in-depth characterization of the lifetime changes at the membrane-substrate interface using cells and in vitro samples that we performed for the revision work.

This characterization of the lifetime changes at the membrane-substrate interface showed a ~0.5ns drop in flipper-TR lifetime in the membranes in contact with non-adherent glass (PLL-g-PEG coated), regardless if the sample is GUVs (Extended data figure 3d), or blebbing, non-adherent cells (extended data figure 8d).

Also, it appears to me that lifetime inside the non-adherent cell is lower than for the bleb's inside. Is this something that the authors see across all samples? If so, how would they justify it?

The lifetime of the membrane in contact with the glass surface is similar whether cells adhere or not (~4.85ns), but non-adhered confined cells show higher flipper-TR lifetime overall as the reviewer pointed, and the 'uropod' or part of the cell behind the bleb tend to show lower lifetime. This might indicate an overall higher membrane tension at blebs, due to confinement and increased hydrostatic pressure, which would also correlate with the increased bleb formation (Sedzinski et al., Nature 2011). The trailing edge enriches in membrane material, ezrin and membrane to cortex attachment proteins (Liu et al., Cell 2015), which might change membrane properties and reduce tension at the trailing edge.

- In Extended Fig.3d, the authors show a lack of lifetime gradient in DOPC/DOPS membranes and hypothesise this is due to the fluid state of the membrane. However, this composition also contains the charged lipid DOPS. Repeating the experiment with pure DOPC GUVs could help to clarify this. Further, I am surprised by the GUV arrangement. DOPC/DOPS is clearly non-spherical, while bSM-containing membranes seem to aggregate. Could the author please comment on this?

We thank the reviewer for inviting us to comment. There are many examples of non-aggregated GUV with ordered compositions and likewise examples with spherical DOPC/DOPC GUVs, but the reviewer is right to comment these trends. It's a common thing to observe GUVs aggregate as on the picture although images shown are usually non-aggregated. This happens because of electrostatic interactions that can occur even with zwitterionic lipids. GUVs containing DOPS have stronger adherence to the glass as shown in the Extended data figure 3 due to increased charges.

We performed some experiments on pure DOPC GUVs and retrieved a similar trend as in the ordered compositions, yet the differences are just of 0.2 ns instead of 0.5 ns.

However, we do not think this new data belongs to the main story of the manuscript and it should be instead in a deeper study about the interaction of pure membrane and substrates, so we propose to not include more data in the manuscript on this topic.

• Fig. 6, HeLa cells seeded on 0.1kPa hydrogels appear not to display a tension gradient. However, I wonder whether this observation could result from an artifact related to the smaller cell size and/or more rounded cell morphology (e.g. more membrane planes are potentially contributing to the FLIM acquisition). Could the authors, try to explore this, e.g. by extracting a decay constant for each of the gradients, and plotting against the cell area?

We thank the reviewer for his/her comments. We do not see a significant difference between the gradient at 0.1 KPa and 0.5kPa despite the clear difference in cell spreading and area, therefore there is no clear correlation between cell area and gradient formation.

These aspects are also addressed by the cytoplasm experiments, which have 20-50% of the cell volume and lack a nucleus but regardless of that still display a lifetime gradient at the plasma membrane interface with the substrate (Extended data figure 2).

We do not think that is therefore useful to include new data or perform extended analysis on this direction.

- Line 134, please clarify which compositions are close to the demixing transition. Provide references and/or experimental data (e.g. DSC) for the melting temperature of these lipid mixtures.

We included two reference to studies using our same three-component mixture (<https://doi.org/10.1021/jp808412x> and Roux et al. EMBOJ 2005). Our GUV data (not shown on the paper for matter of space and focus) agrees with this study (see diagram below). The DOPC:bSM:Chol compositions behave as follows in GUV made with electroformation:

- *DOPC:bSM:Chol 1:1:1, partial separation in domains (see next figure)*
- *DOPC:bSM:Chol 2:2:1, complete separation in domains*
- *DOPC:bSM:Chol 1:1:2, no visible separation in domains.*

Figure Redacted

DOPC:bSM:Chol 1:1:1

	Tau 1	Tau av
Non separated	5.17	4.76
Ld	4.74	4.18
Lo	5.54	5.111

Filename: DOPC-bSM-Chol_1-1-1_003.nd2

We hypothesize that the fully separated 2:2:1 mixture only wets the Ld domain into the glass because it is the most fluid one, as we can see by its low lifetime and behavior in response to the tension gradient, whereas the other two mixtures are able to trigger domain formation in response to tension.

- Line 227, the authors indicate that differences in actin cytoskeleton morphology led to different tension values, as evidenced by different Flipper lifetimes. However, as pointed out by R1, molecular rotors (sensitive to lipid packing) also show a higher lifetime in GUVs in the presence of actin. Because Flipper must also report on lipid packing (given its distinct lifetimes in membranes of different composition), how can the authors rule out the possibility of different lipid packing caused by the attached acting vs a tension gradient in the membrane?

The reviewer is right pointing out that Flipper-TR is a lipid packing probe and that protein adherence to the membrane can influence its viscosity as it was measured by a previous publication using molecular rotors

<https://pubs.acs.org/doi/10.1021/jacsau.4c00237>). However, in the field of cell migration it has been proposed since a few years that membrane to cortex attachment are actually lower at the cell front; meaning that the binding between actin and the membrane is lowest or weakest at the sites of protrusions (reviewed in <https://pubmed.ncbi.nlm.nih.gov/33352139/> but based on work from the Meyer or Danuser labs). The effect that the reviewer is pointing would go in the opposite direction as the data we present in this study. Moreover, it should be hard to reconcile as we point out in the discussion that different actin nucleators have such drastically different effect on packing alone, as they all need to bind to the membrane to activate actin assembly.

- For the MALDI-ToF experiments, Gb3 (a bulky lipid) seems to accumulate at the edges/protrusions, where tension is also higher. The composition of the plasma membrane is believed to be close to the critical points – hence allowing the formation of transient nanodomains. Hence, why Gb3 accumulates at the edges while DOPE-A647 is depleted towards the edge of SLBs corresponding to 1:1:2 and 1:1: DOPC:bSM:Chol composition?

We thank the reviewer for pointing out this interesting aspect: DOPE-atto and Gb3 localize at complementary domains in a phase-separated membrane.

DOPE atto-647 fluorophore is a bulky glycerolipid and will therefore go to places with higher lipid packing defects, in particular Ld phases. As a matter of fact, DOPE-atto lipid analogues are widely used as markers of liquid-disordered phase in vitro (for example this recent publication from the Dimova group in this journal: <https://doi.org/10.1038/s41467-025-57985-2>)

Gb3 is a bulky glycosphingolipid found in lipid rafts, and serves as a specific receptor for the Shiga toxin (Stx). Molecular interactions that mediate lipid raft segregation dominate over steric effects that cause the bulky lipids to move away from ordered phases. Therefore Gb3 prefers to go to Lo phases. In cells, it is generally viewed as a lipid raft maker, therefore indicator of ordered domains. Our MALDI data localizes Gb3(d18:1(4E)/14:1(9Z)) (relatively short acyl chain) at the places with higher Flipper lifetime. Our toxin staining experiments qualitatively agree with this, although the bulk shiga toxin staining show less pronounced localization to the edges. This is consistent with our measurement of higher Flipper-TR lifetime, and lipid packing at the cell edges where branched actin polymerization occurs.

Minor comments:

- Fig. 6d (bleb) – Why does the profile remain flat? From the FLIM image, I would expect an increase in Flipper lifetime at the front's edge (similar to Extended Fig. 8a)

We thank the reviewer for pointing out. We should indeed expect something like that, but we do not appreciate this in the images or in our data analysis. As one can appreciate in the image, the front of stable blebs has very low photon counts as this is moving at velocities on 10 to 30 microns per minute. That means that during our FLIM acquisition (20 seconds) the front can move typically a few microns. During data analysis we sum photons over a lateral window of a few microns which could mask the expected increase within 1-2 microns at the very edge.

- Replace “demixing compositions” (e.g. Line 150) for compositions far away from the transition temperature.

We used all over the results section the term ‘demixing transition’ to specifically qualify the transition from one-phase state to a two-phase state, therefore demixing composition refers to compositions that can undergo a transition from a one-phase state to a

two-phase state, i.e. close to a transition and not far away. These changes were made according to requests of previous reviewers. We inserted a change to clarify the terminology:

Interestingly, compositions far from the demixing transition between the homogenous and segregated states display a lower Flipper-TR lifetime at the front (leading edge) than the rear, closer to the lipid source. In contrast, compositions close to this demixing transition - meaning compositions in which phase separation can be induced - display an inverted lifetime gradient. Thus, considering the gradient from the inner to the outer part of the membrane patch, compositions close to a demixing transition (i.e., demixing compositions) displayed a positive front-rear difference while others displayed a negative front-rear difference (Fig. 1e, right panel).

Based on previous findings²⁸, we hypothesized that the slope inversion in demixing compositions was due to tension-induced phase separation. In this scenario, higher tension segregates the membrane into nanodomains, creating ordered lipid domains, leading to an increased Flipper-TR lifetime. To test this, we used tracer lipids.

- Out of curiosity, why did the authors decide to form SLBs from coated-beads? Why not directly from SUVs, or GUVs? In the later case, the authors would have been able to observe the spread of microscopic Ld and Lo domains.

This technique allows for a big lipid reservoir to spread slowly on the glass surface as a single bilayer, and has been well characterized. This allowed to track the spread of a bilayer over hundreds of microns and over tens of minutes, needed to perform such in depth analysis and overcome the photon count limitations inherent to FLIM. SLBs made from SUVs are subjected to having many pores or discontinuities.